# ADDRESSING DIVERGENT REPRESENTATIONS FROM CAUSAL INTERVENTIONS ON NEURAL NETWORKS

**Satchel Grant**
Stanford University

**Simon Jerome Han**
Stanford University

**Alexa R. Tartaglini**
Stanford University

**Christopher Potts**
Stanford University

## ABSTRACT

A common approach to mechanistic interpretability is to causally manipulate model representations via targeted interventions in order to understand what those representations encode. Here we ask whether such interventions create out-of-distribution (divergent) representations, and whether this raises concerns about how faithful their resulting explanations are to the target model in its natural state. First, we demonstrate theoretically and empirically that common causal intervention techniques often do shift internal representations away from the natural distribution of the target model. Then, we provide a theoretical analysis of two cases of such divergences: 'harmless' divergences that occur in the behavioral null-space of the layer(s) of interest, and 'pernicious' divergences that activate hidden network pathways and cause dormant behavioral changes. Finally, in an effort to mitigate the pernicious cases, we apply and modify the Counterfactual Latent (CL) loss from Grant (2025) allowing representations from causal interventions to remain closer to the natural distribution, reducing the likelihood of harmful divergences while preserving the interpretive power of the interventions. Together, these results highlight a path towards more reliable interpretability methods. [1]

## 1 INTRODUCTION

A central goal of mechanistic interpretability is to understand what the internal representations of neural networks (NNs) encode and how this gives rise to their behavior. Perhaps the most powerful approach to pursuing this goal is through causal interventions, where methods such as activation patching and Distributed Alignment Search (DAS) directly manipulate internal representations to test how they affect outputs (Geiger et al., 2021; 2023; Wu et al., 2023; Wang et al., 2022; Meng et al., 2023; Nanda, 2022; Csordás et al., 2024). Indeed, even correlational methods such as Sparse Autoencoders (SAEs) and Principal Component Analysis (PCA) often use causal interventions as a final judge for whether the features they identify are truly meaningful (Huang et al., 2024; Dai et al., 2024). Causal interventions thus occupy a central place in making functional claims about neural circuitry (Pearl, 2010; Geiger et al., 2024; 2025; Lampinen et al., 2025; Braun et al., 2025).

The use of causal interventions rests on a fundamental assumption that counterfactual model states created by interventions are realistic for the target model. Despite its pervasiveness, however, this assumption is often untested. For example, some activation patching experiments multiply feature values by up to 15x (Lindsey et al., 2025); in these settings, it seems possible that intervened representations diverge significantly from the NN's natural distributions. This raises questions about the reliability of causal interventions for mechanistic interpretability. Do divergent representations change what an intervention can say about an NN's natural mechanisms? When, and to what extent, is it okay for such divergences to occur? When it is not okay, how can we prevent them from occurring?

In this work, we provide both empirical and theoretical insight on these issues. We first demonstrate that divergent representations are a common issue for causal interventions – across a wide range of intervention methods, we find that intervened representations often do diverge from the target NN's natural distribution. We then provide theoretical examples of two types of divergence: 'harmless' divergences that can occur from within-decision-boundary covariance along causal dimensions or from deviations in the null-spaces of the NN layers, and 'pernicious' divergences that activate hidden network pathways and can cause dormant changes to behavior. We provide discussion on

---

[1] Link to public repository: https://github.com/grantsrb/rep_divergence

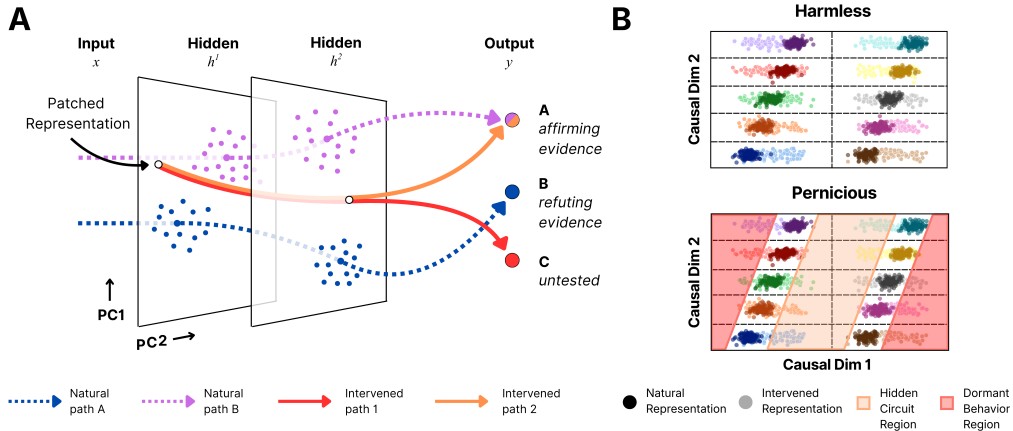

Figure 1: **Causal interventions can recruit hidden circuits that produce misleadingly confirmatory or dormant behavior.** **(a)** Consider natural pathways (dashed arrows) for two classes A and B that carry activity to different behavioral outputs $y$. In a hypothetical intervention meant to find path A, patching $h^1$ with a divergent representation can activate distinct, hidden pathways (solid arrows) that result in misleadingly confirmatory behavior (orange) and/or undetected behavior (red). **(b)** Consider 2D projections of the neural activity of $h^1$ for a different network that classifies states into one of 10 classes (denoted by hue). Suppose that natural representations (dark points) lie within well-defined decision boundaries (dashed lines) and covary along causal axes, and that intervened representations (light points) are constructed by patching the first axis from a sampled natural representation. Although these representations diverge from the natural distribution, this can be harmless (top) or pernicious (bottom) depending on the network's functional landscape. In particular, it can be pernicious if the network has a functional landscape where intervened activity unknowingly recruits hidden circuits (visualized as an orange region) or crosses dormant behavioral boundaries (red regions), depending on the claims.

how harmless and pernicious cases are not always mutually exclusive, where the harm depends on the specific mechanistic claims. Finally, we provide a broad-stroke, initial solution for mitigating pernicious divergences by minimizing all intervened divergences. We show that we can use the Counterfactual Latent (CL) auxiliary loss introduced in Grant (2025) to reduce all representational divergence in the Boundless DAS setting from Wu et al. (2023) while maintaining the same behavioral accuracy; and we introduce a modified version of the CL loss that targets causal subspaces and show that it can improve out-of-distribution (OOD) intervention performance on synthetic tasks. Although we do not propose this method as the final solution to representational divergence, we pose it as a step towards more reliable interventions.

We summarize our contributions as follows:

1. We show theoretical and empirical examples of divergence between natural and causally intervened representations for a variety of causal methods (Section 3).

2. We provide a theoretical treatment of cases in which divergence can arise innocuously from variation in null-spaces, demonstrating that some divergences can even be desired (Section 4.1)

3. We provide synthetic examples of cases where divergent representations can (1) activate hidden computational pathways while still resulting in hypothesis-affirming behavior, and (2) cause dormant behavioral changes, together raising questions about the mechanistic claims that can be made from patching results alone (Section 4.2).

4. Lastly, we use the CL auxiliary loss from Grant (2025) to minimize patching divergence directly in the 7B Large Language Model (LLM) Boundless DAS experimental setting from Wu et al. (2023), and we introduce a stand-alone, modified CL loss that exclusively minimizes divergence along causal dimensions, improving OOD intervention performance in synthetic settings. Together, these results provide an initial step towards mitigating pernicious divergences (Section 5).

## 2 BACKGROUND AND RELATED WORK

### 2.1 ACTIVATION PATCHING

Activation patching generally refers to a process of "patching" (i.e. substituting) some portion of neural activity at an intermediate layer into or from a corrupted forward pass of a network (Geiger et al., 2020; Vig et al., 2020; Wang et al., 2022; Meng et al., 2023; Zhang & Nanda, 2024). It can be performed at various granularities such as whole layers, attention heads, or individual neurons. Many forms of activation patching can be unified under the assumption that subspaces, rather than individual neurons, are the atomic units of NN representations (Rumelhart et al., 1986; McClelland et al., 1986; Smolensky, 1988; Elhage et al., 2022; Geiger et al., 2021; Grant et al., 2024). Activation patching at the level of individual neurons can be understood as subspace patching along neuronal axes, and many of its high-level granularities can be understood as specific forms of individual neuron patching (Geiger et al., 2020; Vig et al., 2020; Wang et al., 2022; Meng et al., 2023).

### 2.2 DISTRIBUTED ALIGNMENT SEARCH

Distributed Alignment Search (DAS) (Geiger et al., 2021; 2023; Wu et al., 2023) can be understood as a form of activation patching that operates in a transformed basis so that specific, causally relevant subspaces can be manipulated analogously to high-level variables from causal abstractions (CAs) (e.g. symbolic programs). Many cases of individual neuron (coordinate) patching can be understood as specific cases of DAS that use the identity transform. We use DAS in Sections 4 and 5, so here we introduce its theory and background.

DAS finds alignments between neural representational subspaces and causal variables from Causal Abstractions (CAs) by testing the hypothesis that an NN's latent state vector $h \in R^{d_m}$ can be transformed into a vector $z \in R^{d_m}$ that consists of orthogonal subspaces encoding interpretable variables. This transformation is performed by a learnable, invertible *Alignment Function* (AF), $z = \mathcal{A}(h)$. We restrict our considerations to linear AFs of the form $\mathcal{A}(h) = Wh$ where $W \in R^{d_m \times d_m}$ is invertible. This transformation allows us to formulate $h$ in terms of interpretable variables and to manipulate encoded values.

For a given CA with variables $\text{var}_i \in \{\text{var}_1, \text{var}_2, ..., \text{var}_n\}$, DAS tests the hypothesis that $z$ is composed of subspaces $\vec{z}_{\text{var}_i} \in R^{d_{\text{var}_i}}$ corresponding to each of the variables from the CA. A causally irrelevant subspace $\vec{z}_{\text{extra}} \in R^{d_{\text{extra}}}$ is also included to encode extraneous, functionally irrelevant neural activity (i.e., the behavioral null-space).

$$\mathcal{A}(h) = z = \begin{bmatrix} \vec{z}_{\text{var}_1} \\ \cdots \\ \vec{z}_{\text{var}_n} \\ \vec{z}_{\text{extra}} \end{bmatrix} \tag{1}$$

Here, each $\vec{z}_{\text{var}_i} \in R^{d_{\text{var}_i}}$ is a column vector of potentially different lengths, where $d_{\text{var}_i}$ is the *subspace size* of $\text{var}_i$, and all subspace sizes satisfy $d_{\text{extra}} + \sum_{i=1}^{n} d_{\text{var}_i} = d_m$. The value of a single causal variable encoded in $h$ can be manipulated through an interchange intervention defined as follows:

$$\hat{h} = \mathcal{A}^{-1}((\mathcal{I} - D_{\text{var}_i})\mathcal{A}(h^{\text{trg}}) + D_{\text{var}_i}\mathcal{A}(h^{\text{src}})) \tag{2}$$

Here, $D_{\text{var}} \in R^{d_m \times d_m}$ is a manually defined, block diagonal, binary matrix that defines the subspace size $d_{\text{var}_i}$, and $\mathcal{I} \in \mathbb{R}^{d_m \times d_m}$ is the identity matrix. Each $D_{\text{var}_i}$ has a set of $d_{\text{var}_i}$ contiguous ones along its diagonal to isolate the dimensions that make up $\vec{z}_{\text{var}_i}$. $h^{\text{src}}$ is the *source vector* from which the subspace activity is harvested, $h^{\text{trg}}$ is the *target vector* into which the harvested activity is patched, and $\hat{h}$ is the resulting intervened vector that replaces $h^{\text{trg}}$ in the model's processing. This allows the model to make predictions using a new value of variable $\text{var}_i$.

To train $\mathcal{A}$, DAS uses *counterfactual behavior* $c \sim \mathcal{D}$ as training labels, where $c$ is generated from the CA. $c$, for a given state of a CA and its context, is the behavior that would have occurred had a causal variable taken a different value and everything else remained the same. $c$ is generated by freezing the state of the environment, changing one or more variable values in the CA, and using the CA to generate new behavior in the same environment using the new values. We train $\mathcal{A}$ on intervention samples while keeping the model parameters frozen, minimizing the following objective

(for non-sequence-based settings):

$$\mathcal{L}_{\text{DAS}}(\mathcal{A}) = -\frac{1}{N} \sum_{k=1}^{N} \log p_{\mathcal{A}}\left(c^{(k)} \,\Big|\, x^{(k)}, \hat{h}^{(k)}\right), \tag{3}$$

where $N$ is the number of samples in the dataset, $c^{(k)}$ is the counterfactual label in sample $k$, $x^{(k)}$ is the model input data, and $p_{\mathcal{A}}(\cdot \mid \cdot)$ is the model's conditional probability distribution given the intervened latent vector, $\hat{h}$. We minimize $\mathcal{L}_{\text{DAS}}(\mathcal{A})$ using gradient descent, backpropagating into $\mathcal{A}$ with all model weights frozen. $\mathcal{A}$ is evaluated on new intervention data, where the model's accuracy on $c$ following each intervention is referred to as the Interchange Intervention Accuracy (IIA).

## 2.3 PROBLEMATIC CAUSAL INTERVENTIONS

Prior work has implicitly explored issues related to representational divergence from causal interventions. For example, methods such as causal scrubbing or noising/denoising activation patching (Wang et al., 2022; LawrenceC et al., 2022; Meng et al., 2023; Chen et al., 2025; Zhang & Nanda, 2024) intentionally introduce divergent representations to test the sufficiency, completeness, and faithfulness of proposed circuits. Works such as Wattenberg & Viégas (2024), Méloux et al. (2024), and Chen et al. (2025) also implicitly explore the dangers of divergent intervened representations by showing how circuits and features can be redundant or have combinatorial effects that are difficult to enumerate given current methodologies, while Zhang & Nanda (2024) and Heimersheim & Nanda (2024) point out easy misinterpretations of patching results. Shi et al. (2024) and Wang et al. (2022) provide criteria centered on faithfulness, completeness, and minimality for evaluating circuits through causal interventions. A body of work on counterfactual explanations exists, some of which has explored differences between on-manifold and off-manifold adversarial attacks (Stutz et al., 2019), and some works have explored constraining counterfactual features to the manifold of the dataset (Verma et al., 2024; Tsiourvas et al., 2024). Our proposed method in Section 5 differs in that it trains a principled alignment to generate counterfactual representations and it constrains deviations along causal dimensions. For DAS in particular, Makelov et al. (2023) demonstrate that it is possible to produce an interaction between the null-space and dormant subspaces that affect behavior. Because they define dormant subspaces as those that do not vary across different model inputs, variation along these directions is, by definition, a form of divergent representation. Finally, Sutter et al. (2025) posit that it is possible to align any causal abstraction to NNs under a number of assumptions including a sufficiently powered, non-linear alignment function, raising questions about what non-linear causal interventions really tell us.

## 3 ARE DIVERGENT REPRESENTATIONS A COMMON PHENOMENON?

We begin by demonstrating that divergent representations are a common (if not likely) outcome of causal interventions, both in theory and in practice. We do not yet consider its perniciousness, however, and we reserve the question of whether and when divergence is harmful for Section 4.

### 3.1 FOR MOST MANIFOLDS, COORDINATE PATCHING GUARANTEES DIVERGENCE

We first consider a theoretical setting where coordinate-based patching of one or more vector dimensions is performed on a single manifold, similar to what might be done in neuron level activation patching (Vig et al., 2020; Geiger et al., 2021). We prove that in this setting, divergent representations are guaranteed to occur if patching is performed exhaustively. For simplicity, we consider a minimal version of this proof that involves a circular manifold with two dimensions. A more general proof, which applies to most manifold geometries, can be found in Appendix A.2.

Formally, let $\mathcal{M}_K = \{\, c_K + u : \|u\|_2 \leq r_K \,\} \subset \mathbb{R}^2$ be a class-$K$ manifold with centroid $c_K \in \mathbb{R}^2$ and radius $r_K > 0$. Given two native representations $h^{src} = c_K + u$ and $h^{trg} = c_K + v$, let us define a coordinate patch (onto class $K$) that keeps the first coordinate of $h^{src}$ and the second of $h^{trg}$:

$$\hat{h} = \begin{bmatrix} h_1^{src} \\ h_2^{trg} \end{bmatrix} = \begin{bmatrix} c_{K,1} + u_1 \\ c_{K,2} + v_2 \end{bmatrix}.$$

Then the deviation from $c_K$ is

$$\hat{h} - c_K = (u_1, v_2)^{\top}, \qquad \|\hat{h} - c_K\|_2^2 = u_1^2 + v_2^2.$$

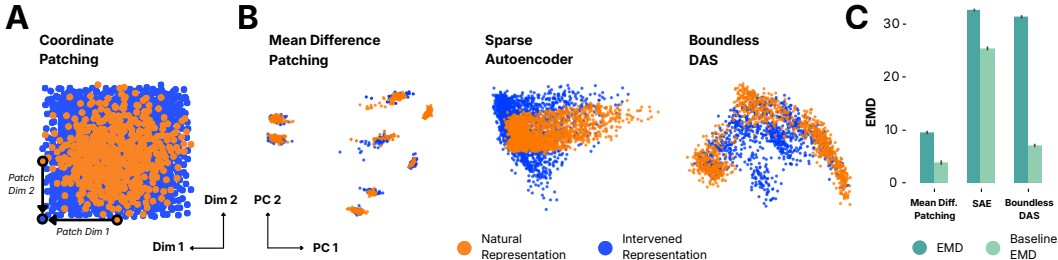

Figure 2: **Representational divergence is a common occurrence across various interventions**. **(a)** Directly replacing a coordinate value in one natural representation (orange) with the value from another will eventually create divergent representations (blue). **(b)** Top two principal components of natural and corresponding intervened representations, taken from the residual stream at the intervention position and with PCA is performed over the combined set of natural and intervened vectors, for three popular causal intervention techniques: a replication of Feng & Steinhardt (2024) for mean difference patching, reconstructed vectors for a single transformer layer using SAELens (Bloom et al., 2024) for sparse autoencoder, and interchange interventions for Boundless DAS (Wu et al., 2023). **(c)** L2 distance between natural and corresponding intervened representations, and Earth Mover's Distance (EMD) between natural and intervened distributions (with baseline comparing the natural distribution to itself).

**Proposition (coordinate patching exceeds the class radius).** If $h^{src}, h^{trg} \in \mathcal{M}_K$ (i.e., $\|u\|_2 \leq r_K$ and $\|v\|_2 \leq r_K$), then the patched point $\hat{h}$ is off-manifold whenever $u_1^2 + v_2^2 > r_K^2$. In particular, there exist boundary points $h^{src}, h^{trg} \in \partial\mathcal{M}_K$ with $u = (r_K, 0)$ and $v = (0, r_K)$ such that $\|\hat{h} - c_K\|_2 = \sqrt{r_K^2 + r_K^2} = r_K\sqrt{2} > r_K$.

**Proof.** Since $\hat{h} - c_K = (u_1, v_2)$, we have $\|\hat{h} - c_K\|_2^2 = u_1^2 + v_2^2$. Choosing $u = (r_K, 0)$ and $v = (0, r_K)$ gives the stated violation. $\qquad\square$

As noted above, this intuition holds for all manifold shapes other than axis-aligned hyper-rectangles (see Appendix A.2). Thus, in these relatively simple theoretical intervention settings, divergent representations are guaranteed to occur with enough intervention samples.

## 3.2 MANY EXISTING CAUSAL METHODS EMPIRICALLY PRODUCE DIVERGENCE

Stepping beyond the theoretical setting, we also empirically demonstrate that common, real-world causal interpretability methods often produce divergent representations. The notion that these methods can create divergence has been indirectly explored in previous work through attention patterns in Gaussian noise corrupted activation patching (Zhang & Nanda, 2024). Here we extend this view to three popular causal intervention methods: mean difference vector patching (e.g. Feng & Steinhardt (2024)), Sparse Autoencoders (e.g. Bloom et al. (2024)), and DAS (e.g. Wu et al. (2023)).

Figure 2 shows the top two principal components of the natural and intervened representations for each intervention method, distinguished by color. These results demonstrate that divergence is a common phenomenon in practice and is not specific to any one method. Even simple methods that patch along a single mean direction are subject to divergence, despite high behavioral accuracy (Feng & Steinhardt, 2024; Geiger et al., 2023; Wu et al., 2023). We quantify this divergence in Figure 2(c) using the Earth Mover's Distance (EMD) (Villani, 2009) between the full dimensional natural and intervened distributions, using the corresponding natural-natural comparison as a baseline. We see that the intervened divergence exceeds that of the natural (more metrics and details are in Appendix A.1). Note that this result does not necessarily imply that the respective methods are invalid or that their claims are incorrect; the panels are only meant to show the presence of divergence.

## 4 WHEN ARE DIVERGENT REPRESENTATIONS HARMLESS OR PERNICIOUS?

Having demonstrated that divergent representations are a common phenomenon, we now consider whether divergence is a *concerning* phenomenon. We propose that divergence is harmless to many

functional claims if it exists in the behavioral null-space, and that it can be pernicious if it recruits hidden pathways or causes dormant behavioral changes. However, we stress that the harm is inherently claim dependent, meaning that these forms of divergence are not mutually exclusive.

## 4.1 Harmless cases

Here we explore a set of cases for which we might consider divergent representations to be harmless to functional claims. First among these are cases in which divergence is bottle-necked into the null-spaces of the next interacting weight matrices. Formally, we define the null-space of a weight matrix $W \in \mathbb{R}^{d' \times d}$ as $\mathcal{N}(W) = \{v \in \mathbb{R}^d \mid Wv = 0\}$. Neural activity in the null-space of the weights refers to any changes $\delta \in \mathbb{R}^d$ for which $W(h + \delta) = Wh$. We propose that divergence $v \in \mathcal{N}(W)$ is harmless to the computation of $W$ because it is equivalent to adding the zero vector $W(h + v) = Wh + Wv = Wh + 0 = W(h + 0)$. Notably, however, this harmlessness does not apply to the sub-computations of the matrix multiplication because $v \in \mathcal{N}(W)$ does not imply that $W_{i,j}(h_j + v_j) = W_{i,j}(h_j + 0)$ for vector row $j$ and matrix row $i$. Thus, $v$ in this case is potentially harmful to mechanistic claims about individual activation-weight sub-computations, while being harmless to the overall matrix multiplication.

We can generalize this notion beyond matrices to an arbitrary function $\psi$. Let $\psi : \mathbb{R}^d \to \mathbb{R}^{d'}$ and let $X \subseteq \mathbb{R}^d$. We define the behavioral null-space with respect to $X$ as

$$\mathcal{N}(\psi, X) = \{ v \in \mathbb{R}^d \mid \forall x \in X, \ \psi(x + v) = \psi(x) \}. \tag{4}$$

A common case of $\psi$ in practice for a layer $\ell$ of an NN $f$ consists of all subsequent computations after and including layer $\ell$, denoted $f_{\geq \ell}(h)$. We propose that behaviorally null divergence $v$ is harmless to the overall computation of $f_{\geq \ell}$ because it is equivalent to adding 0 to the input. However, $v$ can be harmful to claims about a sublayer $\ell + k$ within $f_{\geq \ell}$ because $f_{\ell+k}(f_{\geq \ell, < \ell+k}(h + v))$ is not guaranteed to be equal to $f_{\ell+k}(f_{\geq \ell, < \ell+k}(h + 0))$ (Sec. 4.2.1). See Appx. A.3 and Algorithm 1 to practically classify harmlessness when $\mathcal{N}(\psi, X)$ characterizes the full space of harmless divergence.

**Idealized Case Study.** We now present an example of harmless divergence in the behavioral null-space by introducing a *behaviorally binary subspace*—a vector subspace which causally impacts the outputs of a future processing layer (e.g., classification labels) only through its sign. Formally, let $f : \mathbb{R}^d \to \mathbb{R}^{d'}$ denote a computational unit (possibly consisting of multiple NN layers and functions). Let $\text{sign}(\cdot)$ denote the elementwise sign map $\text{sign} : \mathbb{R}^{d_{\text{var}}} \to \{-1, 1\}^{d_{\text{var}}}$, and assume a fixed alignment function $\mathcal{A}$ and subspace selection matrix $D_{\text{var}} \in \mathbb{R}^{d \times d}$ (Sec. 2.2). A linear subspace $Z \subseteq \mathbb{R}^d$ is *behaviorally binary (with respect to $f$, $D_{var}$, and $\mathcal{A}$)* iff for all $D_{\text{var}}\mathcal{A}(h), D_{\text{var}}\mathcal{A}(h') \in Z$,

$$\text{sign}(D_{\text{var}}\mathcal{A}(h)) = \text{sign}(D_{\text{var}}\mathcal{A}(h')) \implies f(h) = f(h') \tag{5}$$

Now, suppose we have an NN with two causal subspaces, $\tilde{\mathbf{z}}_{\text{var}_\mathbf{a}} \subseteq \mathbb{R}^{d_{\text{var}_a}}$ and $\tilde{\mathbf{z}}_{\text{var}_\mathbf{b}} \subseteq \mathbb{R}^{d_{\text{var}_b}}$, with values $\vec{z}_{\text{var}_a}^{(x_i)}$ and $\vec{z}_{\text{var}_b}^{(x_i)}$, for a model input $x_i$, where the bold, tilde notation distinguishes variables from their values. Further assume that $\tilde{\mathbf{z}}_{\text{var}_\mathbf{b}}$ is behaviorally binary and co-varies with, $\tilde{\mathbf{z}}_{\text{var}_\mathbf{a}}$. Using $h^{(x_i)}$ and $z^{(x_i)}$ from Equation 1 under a given input $x_i$, we use the following definition:

$$\mathcal{A}(h^{(x_i)}) = z^{(x_i)} = \begin{bmatrix} \tilde{\mathbf{z}}_{\text{var}_\mathbf{a}}^{(\mathbf{x_i})} = \vec{z}_{\text{var}_a}^{(x_i)} \\ \tilde{\mathbf{z}}_{\text{var}_\mathbf{b}}^{(\mathbf{x_i})} = \vec{z}_{\text{var}_b}^{(x_i)} \end{bmatrix} \tag{6}$$

Due to the assumption of covariance in $\tilde{\mathbf{z}}_{\text{var}_\mathbf{a}}$ and $\tilde{\mathbf{z}}_{\text{var}_\mathbf{b}}$, it is reasonable to assume that the values $\vec{z}_{\text{var}_b}^{(x_{\text{low}})}$ and $\vec{z}_{\text{var}_b}^{(x_{\text{high}})}$ are systematically distinct for distinct values of $\tilde{\mathbf{z}}_{\text{var}_\mathbf{a}}$ under some classes of inputs, $x_{\text{low}}$ and $x_{\text{high}}$, while $\text{sign}(\vec{z}_{\text{var}_b}^{(x_{\text{low}})}) = \text{sign}(\vec{z}_{\text{var}_b}^{(x_{\text{high}})})$. Now, an interchange intervention on $\tilde{\mathbf{z}}_{\text{var}_\mathbf{b}}$ using a source vector from $x_{\text{low}}$ and target vector from $x_{\text{high}}$, will produce:

$$\hat{z} = \begin{bmatrix} \tilde{\mathbf{z}}_{\text{var}_\mathbf{a}} = \vec{z}_{\text{var}_a}^{(x_{\text{high}})} \\ \tilde{\mathbf{z}}_{\text{var}_\mathbf{b}} = \vec{z}_{\text{var}_b}^{(x_{\text{low}})} \end{bmatrix} \tag{7}$$

Because we assumed that the value of $\vec{z}_{\text{var}_b}^{(x_{\text{low}})}$ is systematically unique due to covariance in $\tilde{\mathbf{z}}_{\text{var}_\mathbf{a}}$ and $\tilde{\mathbf{z}}_{\text{var}_\mathbf{b}}$, the values $\vec{z}_{\text{var}_a}^{(x_{\text{high}})}$ and $\vec{z}_{\text{var}_b}^{(x_{\text{low}})}$ in Equation 7 will have never existed together in the native distribution, but the behavior will remain unchanged because $\tilde{\mathbf{z}}_{\text{var}_\mathbf{b}}$ is behaviorally binary and its sign has not changed. This divergence is thus harmless to the claim of discovering $f$'s causal axes.

**Summary.** We propose that divergences within behavioral null-spaces are harmless to many functional claims about a function $\psi$'s computations when the claim encapsulates (i.e. ignores) the internal, sub-computations of $\psi$. However, we are not suggesting that the behavioral null-space encompasses the set of all harmless divergences. In general, such an exhaustive set cannot exist without assuming the superiority of some scientific claims/assumptions over others. For example, take the set of all harmless divergences for a specific claim, then modify the claim to assume it permissible to diverge in a manner previously excluded from the harmless set. The modification to the claim also modifies the harmless divergences. Lastly, we note that behaviorally null divergence is not always harmless. Indeed, one could even *desire* to intervene on the behavioral null-space to causally test that it is null. Thus, we stress that the mechanistic claim for which an intervention is meant to support is important for determining the harmlessness of the divergence.

## 4.2 PERNICIOUS DIVERGENCE VIA OFF-MANIFOLD ACTIVATION

We now explore pernicious cases of representational divergence involving the concept of *hidden pathways*, which refer to any unit, vector direction, or subcircuit that is inactive on the natural support of representations for a given context but becomes active and influences behavior under an intervention. Although hidden pathways can be compatible with harmless divergences (Sec. 4.1), they can also undermine claims about natural mechanisms and can prime *dormant behavioral changes* discussed in Sec. 4.2.2.

Formally, let $\mathcal{D}$ denote the data distribution over latent representations $h^\ell \in \mathbb{R}^d$ at layer $\ell$, and let $\mathcal{S} = \operatorname{supp}(h^\ell \sim \mathcal{D})$ denote its support with $\mathcal{S}_K = \operatorname{supp}(h^\ell_K \sim \mathcal{D})$ the support for class $K$. Denoting the intended class $K$ following an intervention as subscripted $\to K$, an intervened representation $\hat{h}^\ell_{\to K}$ is said to be *divergent* if $\hat{h}^\ell_{\to K} \notin \mathcal{S}_K$ (e.g. it exists off the natural manifold of class $K$). We define the convex hull of class-K representations as $\operatorname{conv}(\mathcal{S}_K) = \{\sum_i \alpha_i h^\ell_{i,K} : \alpha_i \geq 0, \sum_i \alpha_i = 1\}$ where the subscript $K$ denotes that $h^\ell_{i,K}$ was taken from class $K$ inputs. Projecting an intervention onto $\operatorname{conv}(\mathcal{S}_K)$ ensures that it remains within the convex interpolation region of class-K.

### 4.2.1 MEAN-DIFFERENCE PATCHING CAN ACTIVATE HIDDEN PATHWAYS

Patching with a mean-difference vector can flip a decision by activating a unit that is silent for all natural class inputs.

**Setup.** Consider a two-layer circuit with a ReLU nonlinearity. Let $h^\ell \in \mathbb{R}^4$ feed into

$$s = \mathbf{1}^\top h^{\ell+1} = \mathbf{1}^\top \operatorname{ReLU}(W_\ell h^\ell + b_\ell), \quad W_\ell \in \mathbb{R}^{3\times 4}, \ b_\ell \in \mathbb{R}^3,$$

where

$$W_\ell = \begin{bmatrix} 0.75 & 0.25 & 0 & 0.5 \\ 0 & 1 & 0 & 0 \\ 1 & 1 & -1 & -1 \end{bmatrix}, \quad b_\ell = \begin{bmatrix} -0.5 \\ -0.5 \\ 0 \end{bmatrix}.$$

A positive score ($s > 0$) indicates class A. Suppose class-A and class-B representations at layer $\ell$ are

$$h^\ell_A = \begin{cases} [1,0,1,0]^\top & \text{(case 1)} \\ [0,1,1,0]^\top & \text{(case 2)} \end{cases}, \quad h^\ell_B = \begin{cases} [0,0,1,0]^\top & \text{(case 1)} \\ [0,0,1,1]^\top & \text{(case 2)} \end{cases}$$

Evaluating $h^\ell_A$ yields

$$h^{\ell+1}_{A_{\text{case}_1}} = [0.25, 0, 0]^\top, \quad h^{\ell+1}_{A_{\text{case}_2}} = [0, 0.5, 0]^\top,$$

so for class-A, $s_A \in \{0.25, 0.5\}$. For class-B, all outputs are zero, $s_B = 0$.

**Mean-difference patching.** We construct a mean difference vector between the classes,

$$\delta_{B\to A} = \mu_A - \mu_B = \tfrac{1}{2}\sum_{i=1}^{2}(h^\ell_{A_{\text{case}_i}}) - \tfrac{1}{2}\sum_{i=1}^{2}(h^\ell_{B_{\text{case}_i}}) = [0.5,\, 0.5,\, 0,\, -0.5]^\top.$$

Applying this to class-B representations gives

$$\hat{h}^\ell_{B\to A} = h^\ell_B + \delta_{B\to A} = \begin{cases} [0.5,\, 0.5,\, 1,\, -0.5]^\top & \text{(case 1)} \\ [0.5,\, 0.5,\, 1,\, 0.5]^\top & \text{(case 2)} \end{cases}$$

After propagation through the circuit:

$$\hat{h}^{\ell+1}_{B \to A} = \begin{cases} [0,\, 0,\, 0.5]^\top, & \hat{s}_{\text{case}_1} = 0.5 \\ [0.25,\, 0,\, 0]^\top, & \hat{s}_{\text{case}_2} = 0.25 \end{cases}$$

The intervention flips the decision to class-A ($\hat{s} > 0$). However, the third hidden unit becomes active only for $\hat{h}^\ell_{B \to A}$, never for natural $h^\ell_A$. This new activation is a *hidden pathway* that was silent under all native samples. Thus the mean-difference patch crosses the decision boundary only by activating an off-manifold circuit.

If we project $\hat{h}^\ell_{B \to A}$ onto $\text{conv}(S_A)$ (or equivalently onto a local PCA subspace of $S_A$), this ReLU state change disappears, and the decision boundary is no longer crossed—confirming that the original effect was driven by divergence rather than a within-manifold causal mechanism. It is unclear what this patching experiment reveals about the natural mechanisms of the model.

### 4.2.2 DORMANT BEHAVIORAL CHANGES

Divergent representations can also yield *dormant behavioral changes*: perturbations that appear behaviorally null in one subset of contexts but alter behavior in others. Formally, let $\psi : \mathbb{R}^d \times \mathcal{C} \to \mathbb{R}^{d'}$ and let $\mathcal{C}_1 \subset \mathcal{C}$ be a subset of contexts. The space of dormant behavioral changes relative to $X, \mathcal{C}_1, \mathcal{C}$ is $\mathcal{V}(\psi, X, \mathcal{C}_1, \mathcal{C}) = \mathcal{N}(\psi, X, \mathcal{C}_1) \setminus \mathcal{N}(\psi, X, \mathcal{C})$.

**Illustration.** Extend the network from the previous Sec. 4.2.1 by adding one row of zeros to $W_\ell$ and $b_\ell$, producing $h^{\ell+1} \in \mathbb{R}^4$ where the final coordinate is always zero. Add a context vector $v \in \mathbb{R}^4$ and a final affine layer:

$$\hat{y} = W_{\ell+1}(h^{\ell+1} + v) + b_{\ell+1}, \quad W_{\ell+1} = \begin{bmatrix} 1 & 1 & 0.5 & 0 \\ 0 & 0 & 0 & 0 \\ 0 & 0 & 1 & 1 \end{bmatrix}, \ b_{\ell+1} = \begin{bmatrix} 0 \\ 0.25 \\ -1 \end{bmatrix}.$$

Here, the first argmax index of $\hat{y} = [\hat{y}_1, \hat{y}_2, \hat{y}_3]^\top$ corresponds to class predictions A, B, and C.

Assume $v = [0, 0, 0, v_4]^\top$. For $\hat{h}^{\ell+1}_{B_{\text{case}_1} \to A} = [0, 0, 0.5, 0]^\top$,

$$\hat{y}_1 = 0.25, \quad \hat{y}_2 = 0.25 \quad \hat{y}_3 = (0.5 + v_4) - 1 = v_4 - 0.5.$$

The model predicts class-A when $v_4 < 0.75$ but switches to class-C when $v_4 > 0.75$. Notably, $v_4$ would not naturally cause a class-C prediction below a value of 1 due to the bias threshold. Thus the same intervention that was benign in one context ($v_4 < 0.75$) produces a behavioral flip in another ($1 > v_4 > 0.75$) purely due to the latent divergence priming a new pathway.

**Implications.** Dormant behavioral changes highlight that behaviorally "safe" interventions can still introduce hidden context dependencies. Detecting them would require evaluating across all possible contexts, which is infeasible in practice. Therefore, causal intervention based experiments should ideally (1) report any introduced representational divergence outside of the null-space and (2) test causal interventions for context-sensitivity.

**Summary.** Hidden pathways arise when off-manifold representations activate computations that never occur for natural representations. Such pathways can potentially alter causal conclusions even when behavior *appears* unchanged. Manifold-preserving projections and ReLU-pattern audits are potentially practical safeguards against these pernicious forms of divergence.

## 5 HOW MIGHT WE AVOID DIVERGENT REPRESENTATIONS?

We have thus far shown that divergent representations are common and that their harm depends on multiple factors. We now consider the question of how such divergences might be avoided. Some existing methods solve this by projecting counterfactual features (in our case, intervened representations) directly to the natural manifold (Verma et al., 2024). However, we seek a method that generates principled interventions that are constrained to be innocuous. In this pursuit, we first apply the Counterfactual Latent (CL) loss from Grant (2025) to the DAS experiments from Wu et al. (2023) and find that we can reduce intervened divergence while preserving accuracy on a Llama based LLM. We then introduce a modified CL loss that only targets causal dimensions, and we show that it can improve OOD intervention accuracy. We emphasize, however, that minimizing divergence magnitude does not guarantee elimination of hidden pathways; it only reduces the risk surface.

## 5.1 Applying the Counterfactual Latent Loss to Boundless DAS

To encourage intervened NN representations to be more similar to the native distribution, we first apply the CL auxiliary loss from Grant (2025) to the Boundless DAS setting in Wu et al. (2023). This auxiliary training objective relies on *counterfactual latent (CL) vectors* as vector objectives. CL vectors are defined as vectors that encode the same causal variable value(s) that *would* exist in the intervened vector, $\hat{h}$, assuming the interchange intervention was successful. We can obtain CL vectors from sets of natural $h$ vectors from situations and behaviors that are consistent with the values of the CA to which we are aligning. See Figure 3 for a visualization. As an example, assume that we have a CA with causal variables $\text{var}_u$, $\text{var}_w$, and $\text{var}_{extra}$, and following a causal intervention we expect $\hat{h}$ to have a value of $u$ for variable $\text{var}_u$ and $w$ for variable $\text{var}_w$. A CL vector $h_{CL}$ for this example can be obtained by averaging over a set of $m$ natural representations, $h_{CL} = \frac{1}{m} \sum_{i=1}^{m} h_{CL}^{(x_i)}$, where each $h_{CL}^{(x_i)}$ has the same variable values: $\text{var}_u = u$ and $\text{var}_w = w$ (as labeled by the CA).

The CL auxiliary loss $\mathcal{L}_{CL}$ introduced in Grant (2025) is composed of the mean of an L2 and a cosine distance objective using CL vectors as labels. Using notation defined in Section 2.2, $\mathcal{L}_{CL}$ for a single training sample is defined as follows:

$$\mathcal{L}_{CL}(\hat{h}, h_{CL}) = \frac{1}{2}\|\hat{h} - h_{CL}\|_2^2 - \frac{1}{2}\frac{\hat{h} \cdot h_{CL}}{\|\hat{h}\|_2 \|h_{CL}\|_2} \tag{8}$$

$\mathcal{L}_{CL}$ is combined with the DAS behavioral loss $\mathcal{L}_{DAS}$ into a single loss term using $\epsilon$ as a tunable hyperparameter: $\mathcal{L}_{total} = \epsilon\mathcal{L}_{CL} + \mathcal{L}_{DAS}$. The loss is computed as the mean over batches of samples and optimized using gradient descent (Appendix A.5).

**Results:** We applied the CL loss to the Boundless DAS notebook from Wu et al. (2024) which reproduces the main result from Wu et al. (2023) (see Appendix A.4). Figure 3A provides qualitative visualizations of the decreased divergence when applying the CL loss, and Figure 3B shows both IIA and EMD as a function of increasing the CL weight $\epsilon$. For small values of $\epsilon$, the IIA is maintained (and potentially even improved) while EMD decreases, demonstrating that the CL auxiliary loss can directly reduce divergence in practical interpretability settings without sacrificing the interpretability method.

## 5.2 Modified CL loss improves OOD interventions in synthetic model settings

We next modify the CL loss to work independently of the DAS behavioral loss and show that it improves OOD intervention performance. We simulate an intermediate layer of an NN by constructing a synthetic dataset of $h$ vectors with known feature dimensions and labels $y$ that we use to train a Multi-Layer Perceptron (MLP). The dataset consists of noisy samples around a set of grid points from two feature dimensions with correlational structure. Specifically, we define a set of features as the Cartesian product of two values along the $x_1$-axis $\{-1, 1\}$ and five values along the $x_2$-axis $\{0, 1, 2, 3, 4\}$, resulting in ten unique coordinates that each correspond to one of ten classes. We add noise and covariance to these feature dimensions and concatenate $n$ extra noise dimensions, resulting in simulated vectors $h \in \mathbb{R}^{2+n}$ where $n = 16$ unless otherwise stated. The feature dimensions of these vectors are shown as the natural distributions in Figures 1(b) and 3. We then train a small MLP on these representations to predict the class labels using a standard cross entropy loss. After training, we perform DAS analyses with either the behavioral loss or the CL loss independently. See Appendix A.5 for further details on dataset construction, MLP training, and DAS training.

We modify the CL loss by applying it to individual *causal* subspaces only (as discovered through the DAS training). This allows us to construct $h_{CL}^{\text{var}_i}$ vectors specific to a single causal variable $\text{var}_i$. The modified $\mathcal{L}'_{CL}$ for a single training sample is defined as follows:

$$\hat{h}^{\text{var}_i} = \mathcal{A}^{-1}(D_{\text{var}_i}\mathcal{A}(\hat{h})), \quad h_{CL}^{\text{var}_i} = \text{stopgrad}(\mathcal{A}^{-1}(D_{\text{var}_i}\mathcal{A}(h_{CL}))) \tag{9}$$

$$\mathcal{L}'_{CL} = \sum_{i=1}^{n} \mathcal{L}_{CL}^{\text{var}_i} = \sum_{i=1}^{n} \left( \frac{1}{2}\|\hat{h}^{\text{var}_i} - h_{CL}^{\text{var}_i}\|_2^2 - \frac{1}{2}\frac{\hat{h}^{\text{var}_i} \cdot h_{CL}^{\text{var}_i}}{\|\hat{h}^{\text{var}_i}\|_2 \|h_{CL}^{\text{var}_i}\|_2} \right) \tag{10}$$

**Results:** Figures 3D and 3E provide a qualitative comparison of intervened and native representations for interventions using a trained DAS rotation matrix. Each dot in the figures shows the values of the

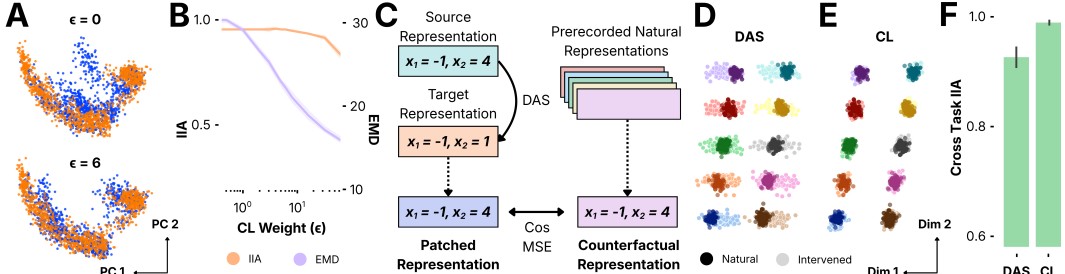

Figure 3: **The CL loss reduces representational divergence and can improve out-of-distribution generalization. (a)** PCA of natural (orange) and intervened (blue) representations in the Boundless DAS setting presented in Wu et al. (2023) for two CL loss weightings with the same final IIA. **(b)** IIA (orange) and divergence (purple) of intervened representations from Section 5.1 as a function of CL loss weight ($\epsilon$). **(c)** Diagram of CL loss; rectangles are model representations and $x_1$ and $x_2$ are deterministic values of the representations along the two synthetic causal dimensions shown in panels (d) and (e). We patch the $x_2$ value from source to target using DAS and define the CL representation as the average of all natural representations that possess the same variable values as the post-intervention representation. **(d)** and **(e)** two causal feature dimensions of representations from a synthetic dataset consisting of ten classes (colors), with both natural (dark) and intervened (light) representations shown. (d) shows results from DAS trained using behavior only, (e) shows DAS trained using only the CL loss. **(f)** performance of alignment matrices trained on one task and evaluated on another that uses the same causal dimensions. CL loss leads to higher OOD performance.

feature dimensions for a single representation. The native states are displayed in darker colors and the intervened in lighter. Each hue indicates the ground truth class of the state. We can see a tightening of the intervened representations using the CL loss. Quantitatively, the DAS loss produces EMD values along the feature dimensions of $0.032 \pm 0.003$ whereas the CL loss produces $0.007 \pm 0.001$ with IIAs of $0.997 \pm 0.001$ and $0.9988 \pm 0.0005$ respectively on training/test sets with held-out classes.

What is the practical utility of reducing divergence? We hypothesized that divergence could influence IIA when transferring the DAS alignment to OOD settings. To test this, we partitioned the synthetic task into a dense and a sparse cluster of classes based on their relative spacing (Appx. A.5.4). We then trained an MLP and alignment on each partition and evaluated the alignment on the held-out partition. The CL loss performed better than the behavioral loss in these OOD settings (Figure 3F). We then regressed OOD IIA on training EMD to find an anti-correlation (coef. -.34, $R^2 = .73$, F(1,28)=75.28, $p < .001$), showing that divergence can predict lower OOD performance (Appx. A.6).

## 6 DISCUSSION AND LIMITATIONS

In this work we demonstrated that a variety of common causal interventions can produce representations that diverge from a target model's natural distribution of latent representations. We then showed that although this can have benign effects for some causal claims, it can also activate hidden pathways and trigger dormant behaviors that can perniciously affect other claims. As a step towards mitigating this issue, we provided a broad-stroke solution by directly minimizing the divergence of intervened activity along causal dimensions, mitigating both pernicious and harmless forms of divergence.

A remaining gap in our work is the failure to produce a principled method for classifying harmful divergence for any claim. Additionally, the modified CL loss is confined to a narrow set of simplistic settings and is not specific to pernicious divergence. We look forward to exploring ways to classify and mitigate pernicious divergence through self-supervised means in future work.

Where does this leave us with respect to causal interventions in mechanistic interpretability? Given our theoretical findings, any divergence outside of the null-space of NN layers is potentially pernicious. This poses challenges for aspirations of a complete mechanistic understanding of NNs using existing methods alone. However, we note that many practical mechanistic projects can be satisfied by collecting sufficiently large intervention evaluation datasets, and continued development of methods such as the CL loss can reduce the problem even further. We are optimistic for the future of this field.

## 7 ACKNOWLEDGMENTS

Thank you to the PDP Lab and the Stanford Psychology department for funding. Thank you to Noah Goodman and Jay McClelland for thoughtful feedback and discussions. Thank you to the PDP lab and the Stanford Mech Interp community for opportunities to present and for thoughtful discussion.

## 8 LLM USAGE STATEMENT

We used ChatGPT to generally edit for clarity, as well as to improve notational consistency in the behavioral null-space, hidden pathways, and dormant behavioral changes formalizations in Sections 4.1 and 4.2. We also used ChatGPT to provide an initial layout of the proof offered in Appendix A.2 showing that axis-aligned hyperrectangles are the only manifold shape that do not have divergent source-target vector pairs in coordinate patching settings, and we used it to suggest, implement, and generate the initial writeup for the additional divergence measures in Appendix A.1.2.

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

## A APPENDIX

### A.1 EMPIRICAL INTERVENED DIVERGENCE METHODOLOGICAL DETAILS

#### A.1.1 INTERVENTION METHODS

We considered three families of interpretability interventions that modify hidden-layer representations. In all three, we visualize the residual stream output from the specified layer:

1. **Mean Difference Vector Patching (MDVP)** (Feng & Steinhardt, 2024), where an intervention vector $\delta_{\text{MD}} \in \mathbb{R}^d$ is defined as the difference in mean activations between two conditions and then added to or subtracted from activations $h \in \mathbb{R}^d$. Formally,

$$\hat{h} = h + \delta_{\text{MD}} \tag{11}$$

   We examine the representations $\hat{h}$ from a sample size of 100 unique contexts across 4 token positions at each individual layer. We compare the representations to the native cases of the swapped binding positions. We used layer 10 of Meta's Meta-Llama-3-8B-Instruct through Huggingface's transformers package for this task and visualization (Touvron et al., 2023; Wolf et al., 2019). We selected layer 10 as it had the lowest EMD difference of all layers, although, we note that this measure did not necessarily correlate with the subjective interpretation of divergence in the qualitative visualizations. We report the EMD difference in Figure 2(a) as the average over all model layers.

2. **Sparse Autoencoder (SAE) Projections** (Bloom et al., 2024), where $h$ is projected through a trained encoder $E : \mathbb{R}^d \to \mathbb{R}^k$ and linear decoder $D : \mathbb{R}^k \to \mathbb{R}^d$:

$$h' = D(E(h)). \tag{12}$$

   SAEs are trained with sparsity penalty $\lambda_{\text{SAE}}$ to encourage interpretable basis functions. We offload further experimental details to the referenced SAElens paper and code base. We compare the reconstructed representations to an equal sample size of 2000 vectors from the natural distribution. We used layer 25 of Meta's Meta-Llama-3-8B-Instruct through Huggingface's transformers package for this task and visualization (Touvron et al., 2023; Wolf et al., 2019). We selected layer 25 as it appeared to be the only layer available through SAElens' pretrained SAEs.

3. **Distributed Alignment Search (DAS)** (Wu et al., 2023), where representations are aligned to a causal abstraction using a learned orthogonal transformation $Q \in \mathbb{R}^{d \times d}$. See Section 2.2 and Wu et al. (2023) for further detail on the method. We compare the intervened representations to an equal sample size of 1000 vectors from the natural distribution. We used the model and layer specified in Wu et al. (2023) for the visualizations in Figure 2.

#### A.1.2 MEASURING DIVERGENCE

For each intervened sample, there exists a corresponding ground truth sample that the intervention is meant to approximate. In the case of the mean difference experiments, these ground samples consist of the naturally occurring entity or attribute in the position which the $\delta_{MD}$ is meant to approximate. For the SAEs, each reconstructed vector has a corresponding encoded vector. For DAS, the ground truth vectors are equivalent to CL vectors.

**Earth Mover's Distance:** To quantify distributional differences between original and intervened representations, we approximated the **Earth Mover's Distance (EMD)** (Villani, 2009) including all vector dimensions using the Sinkhorn loss from the GeomLoss python package with a $p = 2$ and blur $= 0.05$ (Cuturi, 2013). Let $\mathcal{H} = \{h_i\}_{i=1}^N$ denote a set of original representations and $\hat{\mathcal{H}} = \{\hat{h}_i\}_{i=1}^N$ their intervened counterparts. We computed

$$\text{EMD}(\mathcal{H}, \hat{\mathcal{H}}) = \min_{\gamma \in \Pi(\mu, \nu)} \frac{1}{N} \sum_{i,j} \gamma_{ij} \|h_i - \hat{h}_j\|_2, \tag{13}$$

where $N$ is the number of samples in $\mathcal{H}$, $\mu$ and $\nu$ are the empirical distributions over $\mathcal{H}$ and $\hat{\mathcal{H}}$, and $\Pi(\mu, \nu)$ denotes the set of couplings with marginals $\mu$ and $\nu$.

**Baseline divergence.** To ensure that we did not introduce bias from the sampling procedure in the baseline comparison, we use the corresponding ground truth vectors for the intervened vectors when comparing to the baseline divergence. Formally, let $\mathcal{H}'$ be the set of ground truth natural vectors corresponding to the set $\hat{\mathcal{H}}$:

$$\text{Divergence}_{\text{Baseline}} = \text{EMD}(\mathcal{H}, \mathcal{H}'). \tag{14}$$

**Local PCA Distance:** For each reference point $x_i$, we identified its $k$ nearest neighbors $\mathbb{N}_k(x_i)$ in Euclidean space and computed the local tangent subspace via PCA. Let $U_d(x_i) \in \mathbb{R}^{D \times d}$ denote the top $d$ principal components explaining at least 95% of the local variance. The projection matrix onto this tangent space is $P_i = U_d U_d^\top$.

For a query point $v$, the *Local PCA Distance* is the orthogonal residual between $v$ and its projection onto the local tangent space at the nearest reference sample:

$$D_{\text{PCA}}(v) = \min_{x_i} \|(I - P_i)\,[\,v - x_i\,]\|_2\,.$$

Small values indicate that $v$ lies close to the locally linear approximation of the manifold, whereas large residuals reflect departures orthogonal to the manifold surface.

**Local Linear Reconstruction Error:** We computed an error inspired by Locally Linear Embedding (LLE). Given the same neighborhood $\mathcal{N}_k(v)$ of $k$ reference points $X_k = \{x_1, \ldots, x_k\}$, we found reconstruction weights $w = (w_1, \ldots, w_k)$ that best express $v$ as a convex combination of its neighbors:

$$\min_w \left\| v - \sum_{j=1}^{k} w_j x_j \right\|_2^2 \qquad \text{s.t.} \qquad \sum_j w_j = 1.$$

A small Tikhonov regularizer $\lambda I$ was added to the local covariance for numerical stability. The *Local Linear Reconstruction Error* is the residual norm at the optimum:

$$D_{\text{LLR}}(v) = \|v - X_k w^*\|_2\,.$$

This metric measures how well $v$ can be expressed as a locally linear interpolation of nearby manifold points; poor reconstruction (large $D_{\text{LLR}}$) implies off-manifold position or local curvature mismatch.

**Kernel Density (KDE) Density Score:** We estimated a nonparametric probability density function over the reference set using Gaussian kernel density estimation:

$$\hat{p}(x) = \frac{1}{nh^D} \sum_{i=1}^{n} \exp\left[ -\frac{\|x - x_i\|_2^2}{2h^2} \right],$$

where $h$ is the kernel bandwidth determined by Silverman's rule of thumb or cross-validation. The *KDE Density Score* for a query point $v$ is its log-density under this model,

$$S_{\text{KDE}}(v) = \log \hat{p}(v),$$

which inversely reflects off-manifold distance: lower log-density corresponds to less typical or out-of-distribution points. To express results on a comparable scale, we report the negative log-density (i.e., $-S_{\text{KDE}}$), so that larger values consistently indicate greater deviation from the manifold.

### A.1.3 VISUALIZATION

We visualized both original and intervened representations by projecting onto the first two principal components of the covariance matrix of $\mathcal{H}'$ and $\hat{\mathcal{H}}$ combined:

$$\text{PCs} = \text{eigenvectors}\left(\text{Cov}\left(\begin{bmatrix} \mathcal{H}' \\ \hat{\mathcal{H}} \end{bmatrix}\right)\right). \tag{15}$$

The top two principal components were used to plot representations in two dimensions, with colors distinguishing intervention method and condition (Figure 2).

## A.2 Closure under coordinate patching characterizes Cartesian products and hyperrectangles

In this section, we will show that axis-aligned hyperrectangles are the only convex manifold shape that does not have source-target vector pairs that produce off-manifold intervened representations.

For vector dimensions $S \subseteq [d] = \{1, \ldots, d\}$ and $h^{src}, h^{trg} \in \mathbb{R}^d$, define a coordinate patch

$$\text{Patch}_S(h^{src}, h^{trg})_i = \begin{cases} h_i^{src}, & i \in S, \\ h_i^{trg}, & i \notin S. \end{cases}$$

where $i$ refers to the $i^{\text{th}}$ vector coordinate. A set $\mathcal{M} \subseteq \mathbb{R}^d$ is *patch-closed* if $\text{Patch}_S(h^{src}, h^{trg}) \in \mathcal{M}$ for all $h^{src}, h^{trg} \in \mathcal{M}$ and all $S \subseteq [d]$.

Let $\pi_i : \mathbb{R}^d \to \mathbb{R}$ be the $i^{\text{th}}$ coordinate projection and write $I_i := \pi_i(\mathcal{M}) = \{h_i : h \in \mathcal{M}\}$.

**Theorem A.2 (Patch-closure $\iff$ product of projections).** Let $\mathcal{M} \subseteq \mathbb{R}^d$ be nonempty and let $I_i := \pi_i(\mathcal{M})$. Then

$$\mathcal{M} \text{ is patch-closed} \iff \mathcal{M} = I_1 \times \cdots \times I_d.$$

*Proof.* ($\Leftarrow$) Immediate: if $\mathcal{M} = \prod_i I_i$, then patching replaces coordinates by elements of the corresponding $I_i$, so the result stays in $\mathcal{M}$.

($\Rightarrow$) Suppose $\mathcal{M}$ is patch-closed. The inclusion $\mathcal{M} \subseteq \prod_i I_i$ is tautological. For the reverse inclusion, fix $t = (t_1, \ldots, t_d)$ with $t_i \in I_i$. For each $i$ pick $h^{(i)} \in \mathcal{M}$ with $h_i^{(i)} = t_i$. Define $\hat{h}^{(1)} := h^{(1)}$ and for $k \geq 2$ set $\hat{h}^{(k)} := \text{Patch}_{\{k\}}(\hat{h}^{(k-1)}, h^{(k)})$. Patch-closure gives $\hat{h}^{(k)} \in \mathcal{M}$, and by construction $\hat{h}_j^{(k)} = t_j$ for all $j \leq k$. Hence $\hat{h}^{(d)} = t \in \mathcal{M}$, yielding $\prod_i I_i \subseteq \mathcal{M}$. $\qquad\square$

**Corollary (Convex case $\Rightarrow$ hyperrectangle).** If $\mathcal{M}$ is also convex, then each $I_i = \pi_i(\mathcal{M}) \subset \mathbb{R}$ is convex, hence an interval. Therefore $\mathcal{M} = \prod_i I_i$ is an axis-aligned hyperrectangle (Cartesian product of intervals). Conversely, any axis-aligned hyperrectangle is patch-closed (and convex).

**Implication.** Consequently, any nonempty convex set in $\mathbb{R}^d$ that is not an axis-aligned hyperrectangle (e.g., a ball, ellipsoid, or a polytope with non-axis-aligned faces) fails to be patch-closed: there exist $h^{src}, h^{trg} \in \mathcal{M}$ and $S \subseteq [d]$ such that $\text{Patch}_S(h^{src}, h^{trg}) \notin \mathcal{M}$.

### A.2.1 Activation patching in balanced subspaces

Here we include an additional example of pernicious activation patching that assumes the existence of *balanced subspaces*, defined as one or more behaviorally relevant subspaces that are canceled out by opposing weight values. Before continuing, we note that such subspaces are unlikely to exist in practical models due to the fact that they would only arise in cases where two rows of a weight matrix $W \in \mathbb{R}^{n \times m}$ are non-zero, scalar multiples of one another, assuming $h = Wx$. This example, however, could arise in cases where the input $x$ is low rank and a subset of the columns of two rows in $W$ are scalar multiples of one another.

Consider the case where there exists an NN layer that classifies inputs based on the mean intensity of dimensions 3 and 4 for a latent vector $h \in \mathbb{R}^4$, where the NN layers that produce $h$ are denoted $f(x)$ with data inputs $x$ sampled from the dataset, $x \sim \mathcal{D}$, and where the layer of interest has a synthetically constructed weight vector $w \in \mathbb{R}^4$ where $w^\top = [w_1 \quad w_2 \quad w_3 \quad w_4] = [1 \quad -1 \quad \frac{1}{2} \quad \frac{1}{2}]$. The layer is thus defined as follows:

$$y = w^\top f(x^{(i)}) = w^\top h^{(i)} = 1h_1^{(i)} - 1h_2^{(i)} + \frac{1}{2}h_3^{(i)} + \frac{1}{2}h_4^{(i)} \tag{16}$$

Here, $i$ denotes the index of the data within the dataset. Further assume that some behavioral decision depends on the sign of $y$, that $h_1$ and $h_2$ together form balanced subspaces given $w$ (meaning that for all $x^{(i)} \sim \mathcal{D}$, $w_1 h_1^{(i)} = -w_2 h_2^{(i)}$), and that they are non-dormant, meaning that for some pairs $(x^{(i)}, x^{(j)})$ where $i \neq j$, then $h_1^{(i)} \neq h_1^{(j)}$. Under these assumptions, the subspace spanned by $[1 \quad 0 \quad 0 \quad 0]^\top$ and $[0 \quad 1 \quad 0 \quad 0]^\top$ is not causally affecting the network's output under the natural distribution of $h$. However, if we intervene on $h_1$ or $h_2$ while leaving $h_3$ and $h_4$ unchanged, the intervened representation $\hat{h}$ will diverge and potentially cross the decision boundary.

Concretely, if we set $h^{(i)} = [1 \ 1 \ 1 \ 1]^\top$ and $h^{(j)} = [3 \ 3 \ -1 \ -1]^\top$ and then perform an intervention on $h_2$ using $h^{(i)}$ as the target and $h^{(j)}$ as the source, we get: $\hat{h} = [1 \ 3 \ 1 \ 1]^\top$. This will result in a negative value of $y$, thus crossing its decision boundary using a non-native mechanism. This intervention could be used as experimentally affirming evidence for a mechanistic claim, when in reality we have not addressed the model's original mechanism.

### A.3 PRACTICAL ALGORITHM FOR HARMLESS DIVERGENCE WHEN BEHAVIORAL NULL-SPACE CHARACTERIZES THE FULL HARMLESS SET

In settings where perturbations $v \in \mathcal{N}(\psi, X)$ are treated as harmless and perturbations $v \notin \mathcal{N}(\psi, X)$ as harmful, the behavioral null-space formalism suggests a practical procedure for testing the harmlessness of a given divergence $v$. Let $X_K \subset \mathbb{R}^d$ be the set of natural representations for class $K$, and let $\hat{x}_K \in \mathbb{R}^d$ be an intervened representation for class $K$. To approximate the natural manifold $\mathcal{M}_K$ locally around $\hat{x}_K$, we first select the $n$ nearest neighbors of $\hat{x}_K$ in $X_K$:

$$\mathbb{N}_n(\hat{x}_K) = \{x^{(1)}, \dots, x^{(n)}\} \subset X_K.$$

Let $U \in \mathbb{R}^{n \times d}$ be the matrix whose rows are the neighbors $u_i^\top = (x^{(i)})^\top$, and let

$$\mu_K = \frac{1}{n} \sum_{i=1}^n x^{(i)} \in \mathbb{R}^d$$

denote their mean. Define the centered data matrix

$$\tilde{U} = \begin{bmatrix} (x^{(1)} - \mu_K)^\top \\ \vdots \\ (x^{(n)} - \mu_K)^\top \end{bmatrix} \in \mathbb{R}^{n \times d}.$$

We compute a rank-$r$ PCA of $\tilde{U}$, obtaining the top $r$ principal directions $Q_r \in \mathbb{R}^{d \times r}$ (columns are orthonormal). The corresponding local projection operator is

$$\Pi_K(x) = \mu_K + Q_r Q_r^\top (x - \mu_K).$$

The local projection of the intervened representation is then

$$\hat{x}_{\text{proj}} = \Pi_K(\hat{x}_K),$$

and we define the divergence vector as

$$v = \hat{x}_K - \hat{x}_{\text{proj}}.$$

We now provide Algorithm 1 as a practical method for classifying divergence as harmless or pernicious. We note, however, this algorithm only approximates harmlessness and is not guaranteed to be successful. This is especially the case for situations prone to dormant behavioral changes (Sec. 4.2.2).

### A.4 CL LOSS APPLIED TO BOUNDLESS DAS

To perform the Boundless DAS experiments, we used the Boundless DAS tutorial provided in the pyvene python package (Wu et al., 2024) which reproduces a main result from Wu et al. (2023), and we included the CL loss as a weighted auxiliary objective as described in Section 5.1. The exact task used in this tutorial is one involving continuous valued features, which resulted in few occurrences of valid CL vectors in the provided dataset. In order to obtain exact CL vectors, we generated a token sequence sample that contained a valid CL vector for each intervention sample in the dataset. We left hyperparameter choices the same across all trainings except for the CL loss weight $\epsilon$.

### A.5 CL LOSS IN SYNTHETIC SETTINGS

Here we continue Section 5 with experimental details and additional experiments and results.

---

**Algorithm 1** Classifying the harmlessness of a divergence vector when the behavioral null-space characterizes harmlessness

---

**Require:** Intervened representation $\hat{x}_K \in \mathbb{R}^d$ for class $K$; natural class representations $X_K \subset \mathbb{R}^d$; evaluation set $X_{\text{eval}} \subset \mathbb{R}^d$; function $\psi : \mathbb{R}^d \to \mathbb{R}^{d'}$; neighborhood size $n$; local dimension $r$; tolerance $\epsilon \geq 0$.
**Ensure:** Classification of the divergence vector $v$ as harmless or harmful.
 1: (**Local manifold estimation for class** $K$) Let $\mathbb{N}_n(\hat{x}_K) = \{x^{(1)}, \ldots, x^{(n)}\} \subset X_K$ be the $n$ nearest neighbors of $\hat{x}_K$ in $X_K$.
 2: Compute the mean $\mu_K = \dfrac{1}{n} \displaystyle\sum_{i=1}^{n} x^{(i)}$.
 3: Form the centered matrix $\tilde{U} \in \mathbb{R}^{n \times d}$ with rows $(x^{(i)} - \mu_K)^\top$.
 4: Perform rank-$r$ PCA on $\tilde{U}$ to obtain the top $r$ principal directions $Q_r \in \mathbb{R}^{d \times r}$.
 5: Define the local projection $\Pi_K(x) = \mu_K + Q_r Q_r^\top (x - \mu_K)$ and set $\hat{x}_{\text{proj}} \leftarrow \Pi_K(\hat{x}_K)$.
 6: Define the divergence vector $v \leftarrow \hat{x}_K - \hat{x}_{\text{proj}}$.
 7: (**Behavioral test over a broader context**) For each $x \in X_{\text{eval}}$, compute $\Delta(\psi, x) = \|\psi(x + v) - \psi(x)\|$.
 8: **if** $\max_{x \in X_{\text{eval}}} \Delta(x) \leq \epsilon$ **then**
 9:     **return** $v$ is HARMLESS.
10: **else**
11:     **return** $v$ is HARMFUL.
12: **end if**

---

### A.5.1 SYNTHETIC DATASET CONSTRUCTION

The default synthetic task reported in the results section of Section 5.2 was constructed as a dataset of simulated intermediate-layer representations $h \in \mathbb{R}^{18}$ with known ground-truth labels $y \in \{1, \ldots, 10\}$ and two causal feature dimensions, where 18 comes from 2 feature dimensions plus 16 concatenated noise dimensions is the total feature dimensionality. We split these classes into partition 1 and 2, each consisting of 8 of the 10 classes. The held out classes for partition 1 were contained in partition 2 and visa versa. See Figure 5 for a visualization of the task.

**Base feature coordinates.** We first defined a grid of base coordinates as the Cartesian product of $\{-1, 1\}$ along the first feature axis and $\{0, 1, 2, 3, 4\}$ along the second feature axis:

$$\mathcal{G} = \{-1, 1\} \times \{0, 1, 2, 3, 4\}, \tag{17}$$

This procedure yields $10 = |\mathcal{G}|$ unique base coordinates, each corresponding to a distinct class label.

**Noise and correlation structure.** For each base coordinate $(x_1, x_2) \in \mathcal{G}$, we generated $N$ noisy samples by adding Gaussian noise with variance $0.1^2$ and covariance parameter 0.2. Specifically, each sample was drawn as

$$\begin{bmatrix} \tilde{x}_1 \\ \tilde{x}_2 \end{bmatrix} \sim \mathcal{N}\left( \begin{bmatrix} x_1 \\ x_2 \end{bmatrix}, \begin{bmatrix} 0.1^2 & 0.2 \\ 0.2 & 0.1^2 \end{bmatrix} \right). \tag{18}$$

**Additional noise dimensions.** We augmented each 2D noisy base vector with 16 independent Gaussian noise features, each sampled from $\mathcal{N}(0, 1)$, producing final representations $h \in \mathbb{R}^{2+16}$.

### A.5.2 MLP TRAINING

We trained a feedforward Multi-Layer Perceptron (MLP) classifier to predict the class label $y$ from the synthetic representations $h$. The MLP was parameterized with:

- input dimensionality $d$, defaulting to $d = 18$ as previously described
- a 1D batch normalization of $d$ dimensions (Ioffe & Szegedy, 2015),
- one hidden layer of width 128,
- activation function ReLU,
- dropout with probability 0.5 to drop (Srivastava et al., 2014),
- a 1D batch normalization of 128 dimensions,

- output layer with 10 logits.

Training was performed with a standard categorical cross-entropy loss:

$$\mathcal{L}_{\text{CE}} = -\frac{1}{B} \sum_{i=1}^{B} \log p_\theta(y_i \mid h_i), \tag{19}$$

where $B$ is the batch size and $p_\theta$ denotes the MLP's predictive distribution over class labels.

We perform the the MLP training on both partitions combined for the default dataset split. Then we perform an alignment function training on each partition independently and test the alignment on the untrained partition. We report the average IIA over both data partitions for DAS analyses over 5 seeds. Note that we use an independent MLP for each partition in the OOD experiment (Appendix A.5.4.

Optimization used stochastic gradient descent with learning rate $0.01$ for $300$ epochs with early stopping using an Adam optimizer (Kingma & Ba, 2017). The code was implemented in PyTorch.

### A.5.3 DAS TRAINING

Following MLP pretraining, we applied Distributed Alignment Search (DAS) with varying intensities of the behavioral loss and contrastive learning (CL) loss terms. Specifically, the DAS objective was

$$\mathcal{L}_{\text{DAS}} = \epsilon_{\text{behavior}} \mathcal{L}_{\text{behavior}} + \epsilon_{\text{CL}} \mathcal{L}_{\text{CL}}, \tag{20}$$

where $\epsilon_{\text{behavior}}$ and $\epsilon_{\text{CL}}$ are tunable coefficients controlling the strength of each term. We only use values of 0 or 1 for $\epsilon_{\text{behavior}}$ and we explore values of $\epsilon_{\text{CL}}$ referring to it as the CL epsilon in figures. We default to an overall learning rate of $0.01$ and subspace size of $1$ unless otherwise specified. Details of these loss functions are provided in Section 5.

Importantly, we stopped training after loss convergence with a patience of 400 training epochs, and we kept the best DAS alignment matrix decided by IIA, and the best CL alignment matrix by EMD. Furthermore, for these trainings, we used a symmetric invertible linear weight matrix as our alignment function as introduced in Grant et al. (2024). Namely, the linear alignment matrix $X$ is constructed as $X = (MM^\top + \lambda I)S$ where $M \in R^{d_m \times d_m}$ is a matrix of learned parameters initially sampled from a centered gaussian distribution with a standard deviation of $\frac{1}{d_m}$, $I \in R^{d_m \times d_m}$ is the identity matrix, $\lambda = 0.1$ to prevent singular values equal to 0, and $S \in R^{d_m \times d_m}$ is a diagonal matrix to learn a sign for each column of $X$ using diagonal values $s_{i,i} = \text{Tanh}(a_i) + \lambda(\text{sign}(\text{Tanh}(a_i)))$ where each $a_i$ is a learned parameter and $\lambda = 0.1$ to prevent 0 values. We perform the the alignment function training on both partitions for each synthetic dataset and test each DAS alignment on the untrained partition. We report the average IIA over both data partitions for DAS analyses.

### A.5.4 OUT-OF-DISTRIBUTION EXPERIMENTAL DETAILS

To perform the OOD CL loss experiments, we partitioned the classes into 2 non-overlapping groups. Two of the 10 classes were excluded entirely. The groups were chosen so that the Sparse set had strictly greater spacing than the Dense set. See a visualization of the Dense and Sparse partitions in Figure 5. A separate MLP training was performed on each partition individually. Then an alignment function was trained on each partition/classifier tuple using the settings specified in Appendix A.5.3. The alignment functions were then tested on the untrained partition. We report IIA values averaged over the performance on each partition.

### A.5.5 FURTHER CL LOSS EXPLORATIONS

In these explorations, we explore DAS learning rate and the number of extra noisy dimensions for the OOD experiments. We show accuracies, EMD divergences, and EMD divergences restricted to the causal dimensions. The EMD values are scaled by the number of extra noisy dimensions. We refer to the EMD measurements along causal dimensions only as the Row EMD. We do this for both the trained partitions and held-out partitions for various DAS trainings.

### A.6 LINEAR REGRESSION

In an effort to establish a more general, concrete relationship between intervened divergence an out-of-distribution (OOD) intervention performance, we performed a linear regression on trained

alignments using EMD along causal axes (as discovered through the alignment training) as the independent variable and interchange intervention accuracy (IIA) as the dependent variable. We performed these regressions independently on the Default task and the OOD task trainings, each training consisting of two partitions with 5training seeds and 3 types of alignment trainings: DAS behavioral loss only, CL loss only, and DAS+CL loss, creating 30trainings total. We used the statsmodels python package (Seabold & Perktold, 2010) to perform the regression.

| Dep. Variable: | IIA | R-squared: | 0.729 |
|---|---|---|---|
| Model: | OLS | Adj. R-squared: | 0.719 |
| Method: | Least Squares | F-statistic: | 75.28 |
| Date: | Wed, 26 Nov 2025 | Prob (F-statistic): | 2.00e-09 |
| Time: | 11:54:42 | Log-Likelihood: | 76.575 |
| No. Observations: | 30 | AIC: | -149.2 |
| Df Residuals: | 28 | BIC: | -146.3 |
| Df Model: | 1 | | |
| Covariance Type: | nonrobust | | |

| | coef | std err | t | P> \|t\| | [0.025 | 0.975] |
|---|---|---|---|---|---|---|
| Intercept | 0.9885 | 0.004 | 243.666 | 0.000 | 0.980 | 0.997 |
| Training EMD | -0.3424 | 0.039 | -8.677 | 0.000 | -0.423 | -0.262 |

| Omnibus: | 33.330 | Durbin-Watson: | 1.903 |
|---|---|---|---|
| Prob(Omnibus): | 0.000 | Jarque-Bera (JB): | 98.629 |
| Skew: | -2.235 | Prob(JB): | 3.83e-22 |
| Kurtosis: | 10.676 | Cond. No. | 11.1 |

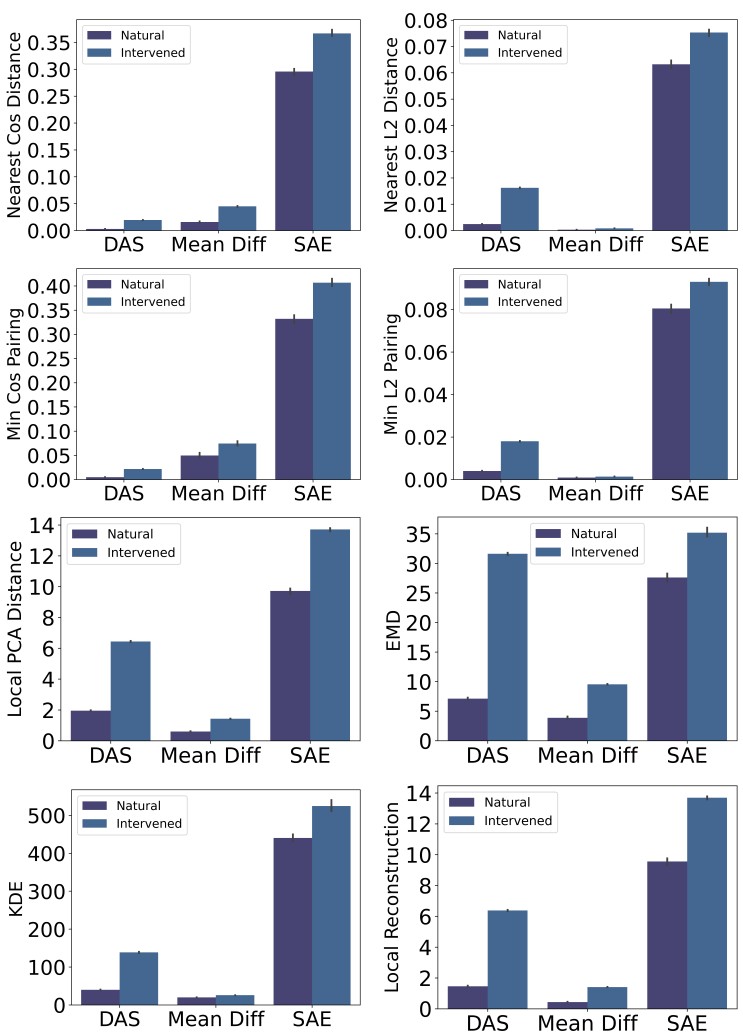

Figure 4: A number of additional divergence measures to demonstrate the difference between the natural and intervened distributions. Each is labeled by its y-axis. Each metric is computed over a random sample of natural vectors to simulate the natural manifold, and a sampled set of intervened or natural vectors for which to measure the distance from the natural distribution. We refer to this distribution as the "compared" distribution. The sampled intervened and natural vectors are always the "ground-truth pair" described at the beginning of Appendix A.1. **Nearest Cosine Distance:** refers to the cosine distance to the nearest sample in the natural manifold. Multiple sampes in the compared distribution can share the same natural sample. This value is averaged over all compared samples. **Nearest L2 Distance** refers to the cosine distance to the nearest sample in the natural manifold. Multiple sampes in the compared distribution can share the same natural sample. This value is averaged over all compared samples. **Min Cos Pairing** refers to the lowest cost pairing where cost is the cosine distance between two samples. Vector pairs are exclusive. This value is normalized by the number of samples. **Min L2 Pairing** refers to the lowest cost pairing where cost is the L2 distance between two samples. Vector pairs are exclusive. This value is normalized by the number of samples. **Local PCA Distance** refers to the distance to the manifold created using a local PCA of the nearest neighbors. See Appendix A.1.2. **EMD** refers to the Earth Mover's Distance. See Appendix A.1.2. **KDE** refers to the Kernel Density Estimation score. See Appendix A.1.2. **Local Linear Reconstruction** refers to the local linear reconstruction error. See Appendix A.1.2.

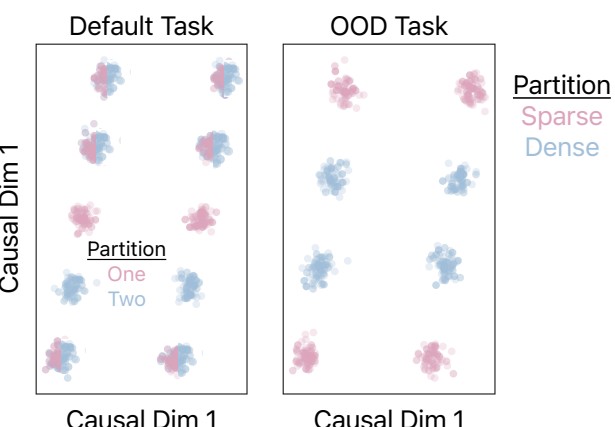

Figure 5: Visualization of the different synthetic tasks used for Figure 3. The Default Task is split into two partitions, both withholding two classes that are contained in the other partition. The OOD task is also split into two partitions, both consisting of 4 classes. The Dense partition consists of a tighter cluster than the Sparse.

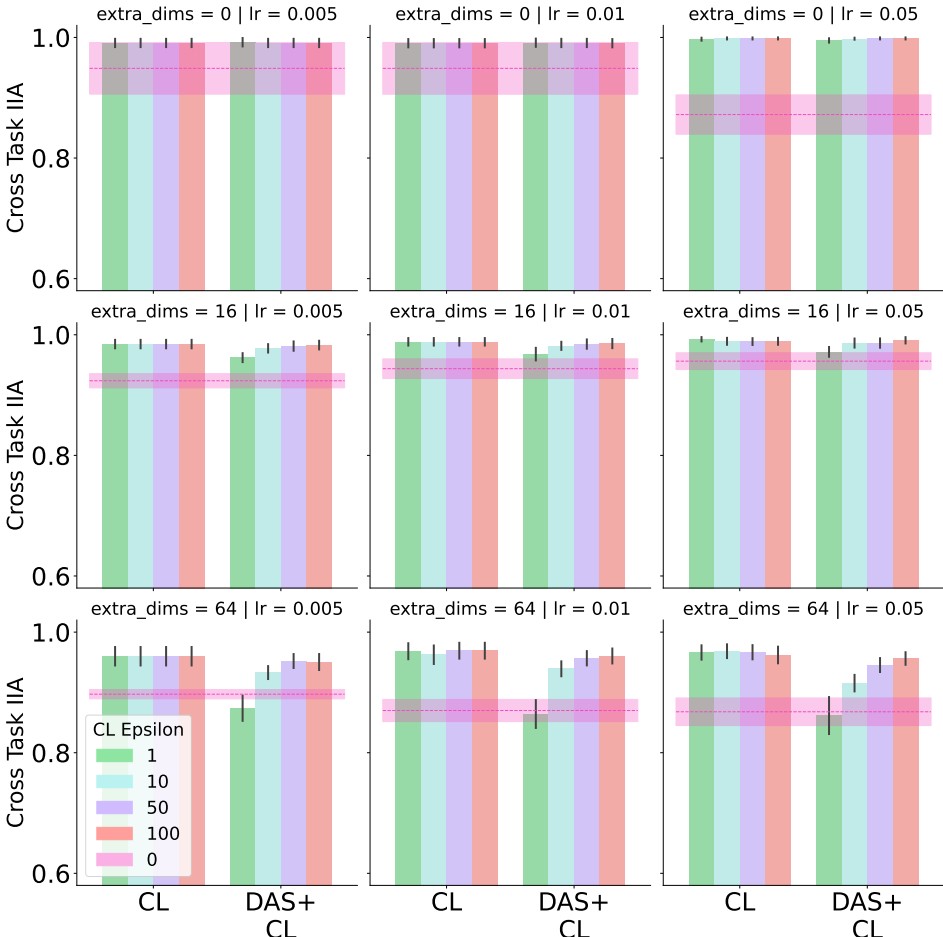

Figure 6: Out of distribution hyperparameter search showing the DAS IIA on OOD validation data for the trained task partition. We see the DAS loss learning rate (lr) and extra concatenated noisy input dimensions (extra_dim) across the panel columns and rows. The DAS+CL reported values include the behavioral loss whereas the CL label excludes the behavioral loss. The pink dashed lines represent DAS trained with the behavioral loss only.

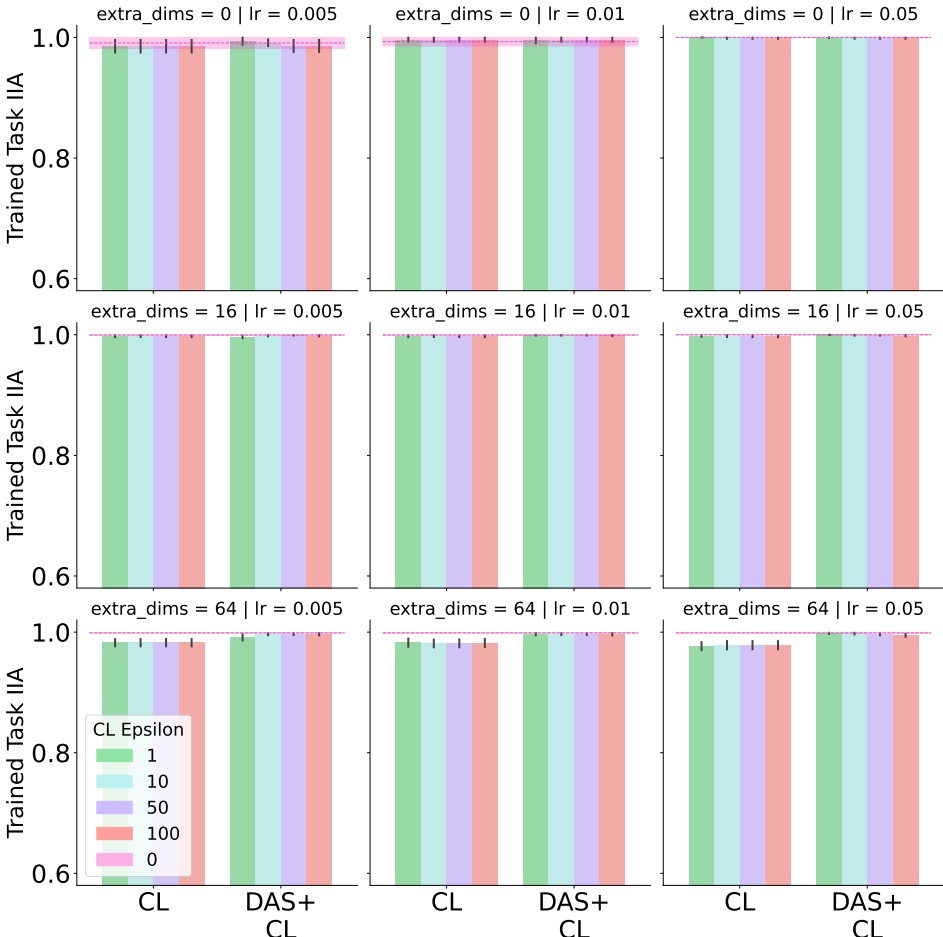

Figure 7: In distribution hyperparameter search showing the DAS IIA on in-distribution validation data for the trained task partition. We see the DAS loss learning rate (lr) and extra concatenated noisy input dimensions (extra_dim) across the panel columns and rows. The DAS+CL reported values include the behavioral loss whereas the CL label excludes the behavioral loss. The pink dashed lines represent DAS trained with the behavioral loss only.

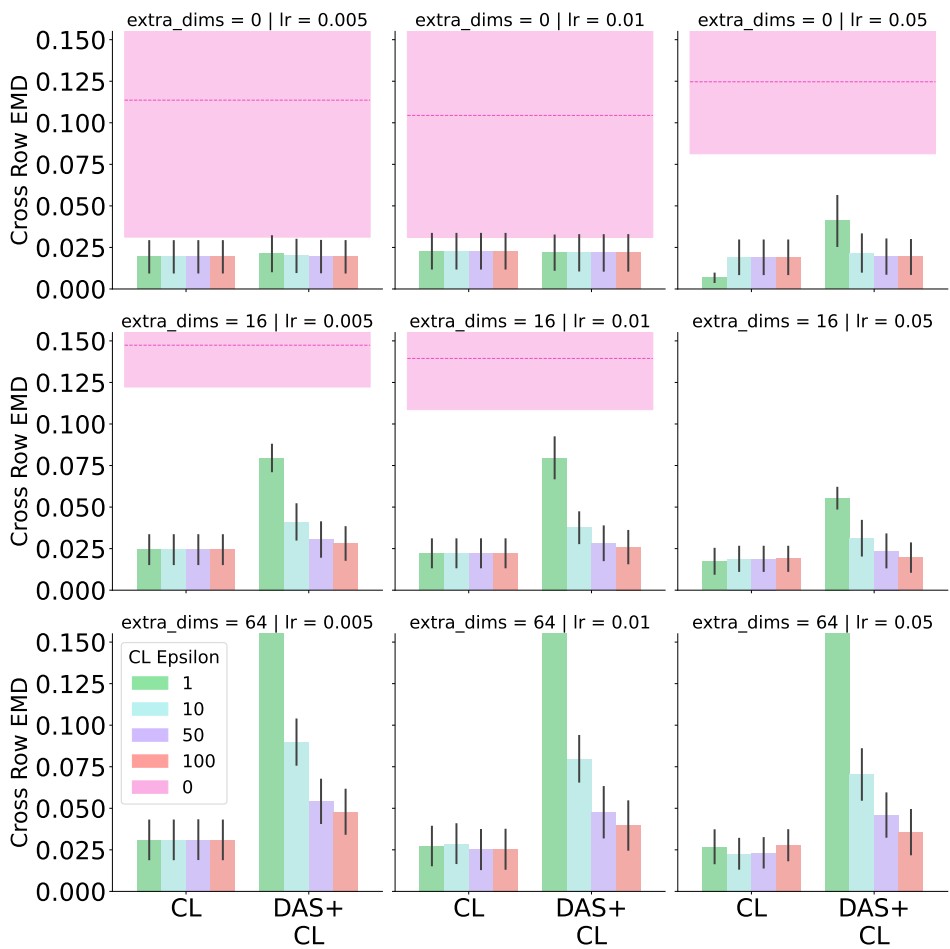

Figure 8: Out of distribution hyperparameter search showing the DAS row-space EMD on OOD validation data for the trained task partition. We see the DAS loss learning rate (lr) and extra concatenated noisy input dimensions (extra_dim) across the panel columns and rows. The DAS+CL reported values include the behavioral loss whereas the CL label excludes the behavioral loss. The pink dashed lines represent DAS trained with the behavioral loss only.

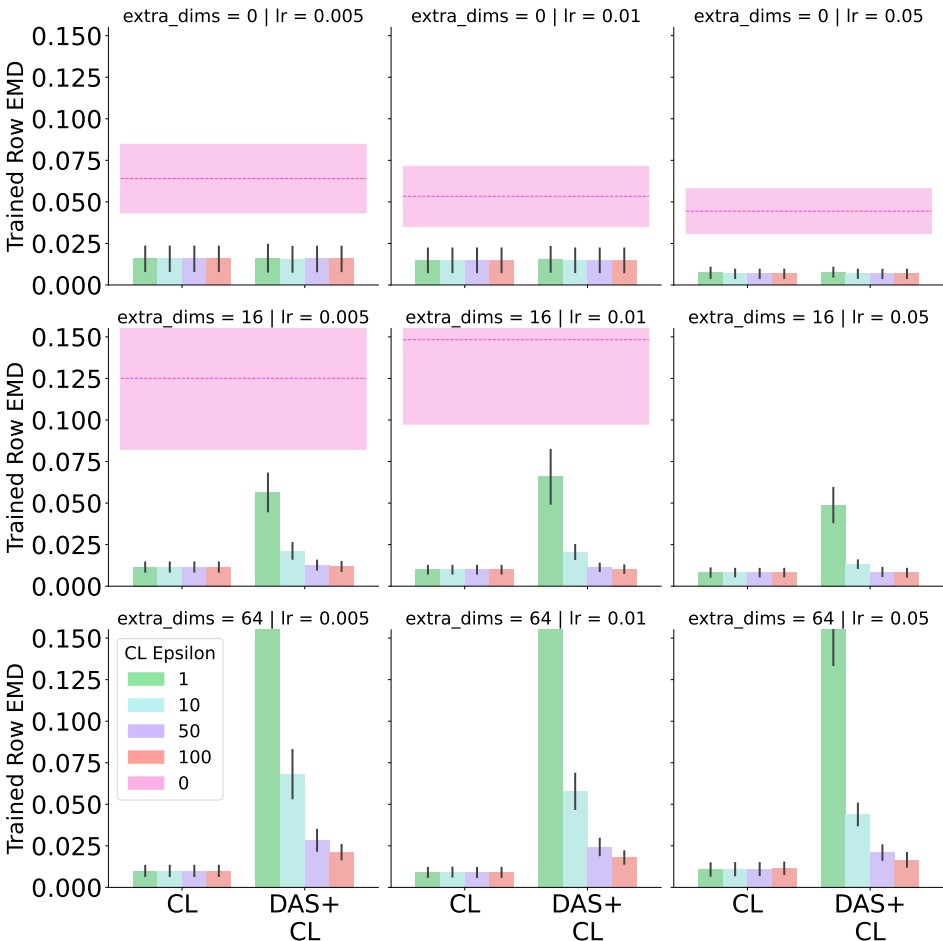

Figure 9: In distribution hyperparameter search showing the DAS row-space EMD on validation data for the trained task partition. We see the DAS loss learning rate (lr) and extra concatenated noisy input dimensions (extra_dim) across the panel columns and rows. The DAS+CL reported values include the behavioral loss whereas the CL label excludes the behavioral loss. The pink dashed lines represent DAS trained with the behavioral loss only.

