# OpenReview forum: "Addressing divergent representations from causal interventions on neural networks"
_ICLR.cc/2026/Conference — ICLR 2026 Oral_

### Official Review · Reviewer_Ahsj · 2025-11-01

**Soundness:** 3
**Presentation:** 3
**Contribution:** 3
**Rating:** 6
**Confidence:** 4

**Summary:**

This paper shows in synthetic settings how causal intervetions, a widely used tool in interpretability, may introduce divergent representations in NNs. The authors studies 2 kinds of such divergences: "harmless", which are in the null space of successive projections; and "pernicious" that may active unintended behaviors downstream. The authors then propose an approach training DAS projections ([Geiger et al, 2023](https://arxiv.org/abs/2303.02536)) incorporating Counterfactual Latent loss ([Grant, 2025](https://arxiv.org/pdf/2501.06164)) to get rid of pernicious divergences.

**Strengths:**

* Solid theoretical discussion on how divergences may arise from causal interventions.
* Great presentation and framing of the problem.

**Weaknesses:**

* Simulated in scenarios with very strong assumptions which may not hold in practice.
* The proposed approach seems unrealistically expensive to be applied in practice.

**Questions:**

1. Figure 2: You show divergences introduced by adding mean diff, patching SAE and DAS. But what about activation patching from a single source sample ([Prakash et al, 2025](https://arxiv.org/abs/2505.14685), [Feucht et al, 2025](https://arxiv.org/pdf/2504.03022), inter alia)? Do you assume such interventions to also introduce divergences?

2. Section 4.2.2: Very interesting observation. Unless I am understanding it wrong, a layernorm right after the intervention can potentially fix this issue, right? You probably should mention this point as most architectures have them nowadays.

3. The situations simulated showing harmless and pernicious divergences are plausible and I do think some of the inteventions will be susceptible to such divergences. However, I am not entirely convinced about how prevalent pernicious divergences are in practice. And usually it is considered a good scientific practice to evaluate proposed hypotheses on 100s (if not 1000s) of examples covering a distribution of scenarios. Do you assume such cases may hold across entire evaluation sets?

4. (related to previous question) Training DAS projections to address this seem quite expensive, even might be just infeasible in practice. The setup makes sense and is convincing in synthetic settings. But do you see any way this can be scaled to real models and datasets? And is it worth the effort [referring to the previous question]? I will be just blunt: is this paper more of an intellectual exercise? Or do you see any practical applications?

---

> ### Author Response · Authors · 2025-11-22
> **Response to weaknesses**
>
> We thank the reviewer for taking the time to provide a thoughtful and constructive review.
>
> In response to the stated weaknesses:
>
> **W1. Simulated in scenarios with very strong assumptions which may not hold in practice.**
>
> Yes, thank you for this opportunity for clarification.
>
> If the concern refers to the balanced subspaces and mean-difference patching sections, we have addressed this in two ways:
>
> - Balanced subspaces: We have moved this material to the supplement since it relies on strong assumptions.
>
> - Mean-difference patching (Section 4.2): The purpose of this example is not to claim realism of the assumptions, but to provide a mechanistic illustration of how representational divergence can arise in this class of interventions. Although it relies on exact values, it clarifies the causal structure underlying mean-difference patching.
>
> If the concern instead refers to Section 3, we have added a new formal result (Section 3.1) showing that for nearly all manifold shapes, activation patching can produce off-manifold intervened representations for some source–target vector pairs. Section 3.2 further demonstrates empirically that divergence is common across several widely used intervention methods.
>
> If the concern refers to the CL loss, we have added an experiment applying CL to an LLM using the price-tagging setup from Wu et al. (2023) (Section 5.1). In this more realistic setting, the CL loss reduces divergence while maintaining interpretability accuracy (updated Figure 3).
>
> **W2. The proposed approach seems unrealistically expensive to be applied in practice.**
>
> As noted above, the updated Section 5.1 demonstrates the practical application of CL to a 7B-parameter LLM. This shows that the method can be implemented efficiently in a realistic experiment, without prohibitive computational cost.
>
> We also emphasize that the CL loss constitutes only a small component of the paper and is intended as an initial mitigating tool rather than a comprehensive solution.

---

> > ### Author Response · Authors · 2025-11-22
> > **Response to questions**
> >
> > **Q1. Figure 2: You show divergences introduced by adding mean diff, patching SAE and DAS. But what about activation patching from a single source sample (Prakash et al, 2025, Feucht et al, 2025, inter alia)? Do you assume such interventions to also introduce divergences?**
> >
> > If we interpret the question correctly, you are asking whether single-sample, vanilla patching (as in Prakash et al. 2025; Feucht et al. 2025) can produce off-manifold representations. We speculate that yes, even such patching can introduce representational divergence beyond the embedding layer.
> >
> > Intuitively, the entire transformer context defines a single point in a manifold of natural contexts. Patching any component of that context effectively alters one coordinate of this high-dimensional manifold point, and this can push the resulting representation off-manifold (see updated Section 3.1). This only applies past the embedding layer, however (see Section 3.1). While we do not provide empirical evidence for this specific case, we believe the intuition is sound.
> >
> > **Q2. Could a layernorm after the intervention prevent the divergence shown in Section 4.2.2?**
> >
> > This is an excellent point. LayerNorm does indeed complicate the use of ReLU activation changes as a simple diagnostic of off-manifold activity. However, LayerNorm does not eliminate the broader notion of representational divergence:
> >
> > - Divergent pre-LN activations change both the mean and variance used by LayerNorm.
> > - Post-LN representations can still deviate from the natural activation manifold.
> > - The example with ReLU is meant as an intuitive illustration, not a comprehensive definition.
> >
> > Thus, even with LayerNorm, intervened representations may diverge from the natural distribution.
> >
> > **Q3. Prevalence of pernicious divergence**
> >
> > How often pernicious divergences arise is likely to depend on the specific experimental setup. For dormant behavioral changes (Section 4.2.2), evaluating interventions on large and diverse evaluation sets is the most practical safeguard that we can provide.
> >
> > For hidden pathway recruitment (Section 4.2.1), we currently lack a principled method to distinguish harmful vs. benign cases without conditioning on a specific causal claim. However, the formal result in updated Section 3.1 suggests that divergence can persist across entire evaluation sets—-although whether that divergence is harmful depends on the causal question being asked.
> >
> > **Q4. Practicality and value of DAS projections—intellectual exercise or practical contribution?**
> >
> > We appreciate the candid framing of this question, and we think it's a good one. As noted earlier, our updated Section 5.1 demonstrates that DAS with CL can be applied directly to an LLM in a realistic experiment (the price-tagging task of Wu et al. 2023). This serves as a proof-of-principle that divergence can be mitigated in practice.
> >
> > More broadly, we view Sections 3 and 4 as making an important conceptual contribution: they highlight that LLM computational trajectories often diverge from the natural manifold yet still function coherently. The exact manner in which this divergence occurs is revealing for the inner workings of neural networks. We also found this divergence surprising and befuddling and, we believe, this topic will be similarly recieved and practically relevant to interpretability researchers.
> >
> > Given the increasing use of causal intervention methods in mechanistic interpretability, we think it is timely to identify and analyze representational divergence as a cross-cutting issue. The CL loss is offered as an initial mitigation strategy, but we view the primary contribution as diagnosing and illustrating the phenomenon itself, which we expect will inform future methodological developments.

---

> > > ### Comment · Reviewer_Ahsj · 2025-11-26
> > >
> > > I thank the authors for their responses and clarifications. To be honest, I am still a bit unsure about the practical implications of this work: how prevalent pernicious divergences are in practice, and to what extent the proposed approach can be scaled to real models and datasets. However, I do think this paper has some very interesting theoretical discussions. I feel that I have learned something new by reading this paper, and it is worth sharing with the community. Therefore, I raise my score to accept.
> > >
> > > I wish the authors good luck with their future research!

---

### Official Review · Reviewer_YbiP · 2025-11-02

**Soundness:** 2
**Presentation:** 2
**Contribution:** 2
**Rating:** 4
**Confidence:** 2

**Summary:**

This paper investigates a critical, often-overlooked assumption in mechanistic interpretability: that causal interventions (like activation patching) produce latent representations that are ”faithful” or ”in-distribution” for the
target model. The authors argue that interventions can create ”divergent,” out-of-distribution states, which may
undermine the validity of the resulting explanations.
The paper’s contributions are threefold:

- It empirically demonstrates that common intervention methods, including Mean Difference Vector Patching, Sparse Autoencoders (SAEs), and Distributed Alignment Search (DAS), do create representations that
diverge significantly from the model’s natural distribution.

-  It provides a theoretical framework for classifying these divergences into ”harmless” (e.g., in the null-space
or within existing decision boundaries) and ”pernicious” (e.g., activating hidden, non-native computational
pathways or causing dormant behavioral changes). The authors use synthetic examples to show how pernicious divergences can lead to misleadingly ”affirming” results.

- To mitigate this issue, the authors propose an adapted Counterfactual Latent (CL) loss (based on Grant
(2025)). This loss regularizes interventions by encouraging the intervened latent state to remain close to an
average of native states that share the same intended causal properties . In a synthetic setting, they show this
method reduces divergence and improves out-of-distribution (OOD) generalization.

**Strengths:**

-  By questioning the
”faithfulness” of intervened states, the work highlights a critical potential failure mode for many mechanistic
interpretability claims.
•  The theoretical categorization of divergences into ”harmless” (e.g., nullspace, within-boundary covariance) and ”pernicious” (e.g., hidden pathways, dormant behavior) is clear,
intuitive, and  valuable.
-  The synthetic examples in Section 4.2 are particularly strong. They offer
concrete, simple-to-understand illustrations of how an intervention can be misleading; for instance, by
breaking a ”balanced subspace” or activating a pathway via a mean-difference vector that is not used by
any native data point . The concept of ”dormant behavioral changes” is also a very insightful and worrying
failure mode.
-  It also proposes a concrete (if
preliminary) mitigation strategy. The adaptation of the CL loss to ”anchor” interventions to the native data
manifold is a good approach.

**Weaknesses:**

- The paper successfully categorizes divergences but does not offer a method
to detect or classify whether a divergence observed in a practical (non-synthetic) setting is harmless or
pernicious. This makes it difficult to know when to be concerned about the results of their
interventions.
-  The primary weakness of the proposed solution, acknowledged by the authors,
is that the CL loss is a ”broad-stroke” approach. It penalizes all divergence, rather than selectively targeting only ”pernicious” divergence. This might be overly restrictive, as some ”harmless” divergences (e.g.,
interventions that explore the null-space) could be desirable for making certain causal claims.

- The empirical validation for the CL loss mitigation is confined to a synthetic, small-scale MLP setting. It is unclear how this approach would scale to modern,
large-scale models. For example, generating the ”Counterfactual Latent” (CL) vectors requires averaging
over a pre-recorded set of native states with specific causal properties. This seems computationally challenging and, more importantly, dependent on having a correct, known causal abstraction, which is often
what is being searched for in the first place.

**Questions:**

-  The CL loss mitigation seems promising but raises practical questions. How do you
envision applying this to large language models where the ground-truth causal variables (needed to find
matching native states for $h_{CL}$) are not known a priori? Doesn’t this create a circular dependency where
you need the causal abstraction to find the CL vectors, but the intervention (which you’re trying to fix) is
what you use to find the abstraction?

- You note that the CL loss is a ”broad-stroke” solution. Do you
have any initial thoughts on how one might distinguish pernicious from harmless divergence, perhaps without a full causal abstraction? For instance, could one use a measure of ”local faithfulness” or analyze
activation changes in all output dimensions (as hinted at in 4.2.3 ) to flag interventions that recruit ”hidden”
pathways?

- The concept of ”dormant behavioral changes” (Section 4.2.3, Appendix A.2) is particularly concerning. You mention that an ”infinitely expansive” dataset could detect them. In practice, how
could a researcher gain any confidence that their intervention hasn’t created such a dormant vulnerability?
Does the CL loss’s reduction in OOD error (Fig 3d) suggest it is mitigating these, or is that a separate issue?

---

> ### Author Response · Authors · 2025-11-22
> **Response to assessment**
>
> We thank the reviewer for taking the time to provide their thoughtful and constructive review.
>
> **W1. No method to determine whether a divergence is harmless or pernicious**
>
> We agree that providing a universal method to classify divergences as "harmless" or “pernicious" is challenging. As we note on line 053, “harmless and pernicious cases are not always mutually exclusive… the harm often depends on the specific mechanistic claims.” Similarly, line 341 emphasizes that “hidden pathways can be compatible with harmless divergences.”
>
> The core difficulty is that harmfulness is inherently claim-dependent: whether divergence undermines an interpretability result depends on the type of causal claim being made and the structure of the hypothesis being tested. Because interpretability studies differ widely in their causal assumptions, we believe that a one-size-fits-all classifier is unlikely to exist.
>
> In settings where divergence is undesirable for the causal claim at hand, the proposed Counterfactual Latent (CL) methodology provides a practical tool for mitigating it.
>
>
> **W2. CL loss is a broad-stroke approach that penalizes all divergence**
>
> We agree. The CL loss is not intended to be universally applicable; rather, it is appropriate only in settings where divergence itself threatens the validity of the intended causal claim. In these cases, minimizing divergence is a feature rather than a limitation.
>
> We also note that the modified CL loss does not uniformly suppress all variation: it specifically targets divergence along causal axes, and we show empirically that this improves out-of-distribution (OOD) intervention accuracy.
>
> **W3. Synthetic, small-scale validation and scalability concerns**
>
> To address this concern, we have added Section 5.1 which provides new experiments on a Llama-based 7B model using the price-tagging setting introduced in Wu et al. (2023). The updated Figure 3 demonstrates that the CL loss reduces divergence while preserving interpretability accuracy in this practical setting.
>
> Regarding practicality: in many real-world interpretability studies, researchers can generate data that contain the appropriate causal variable configurations. We clarify this on lines 930–931: “To obtain exact CL vectors, we generated a token sequence sample that contained a valid CL vector for each intervention sample.”
>
> More broadly, we agree that CL and other DAS-like methods are limited when the goal is automated discovery of causal abstractions. These methods instead support hypothesis-driven interpretability, where the researcher formulates a testable causal abstraction and uses interventions to evaluate it. Our intention is to highlight representational divergence as a cross-cutting issue affecting many causal interpretability methods; the CL loss is provided as an initial step toward mitigating these issues, not as a universal solution.
>
>
>
>
> **Q1. Circularity: CL vectors require known causal variables**
>
> DAS-like methods—including CL—are explicitly hypothesis-driven. They require the experimentalist to propose a causal abstraction that can then be tested. This avoids circularity: the abstraction is posited first, and the intervention is used to evaluate its validity.
>
> For a concrete example of how this is done in practice, please see the updated Section 5.
>
> **Q2. Distinguishing pernicious from harmless divergence without a full causal model**
>
> This is an important open question. Although we do not provide a definitive solution, we outline two practical approaches:
>
> - Detecting hidden pathway recruitment: One can approximate the natural activation manifold at each post-intervention layer using a dataset and then measure the distance of intervened representations to that manifold. While not a guarantee, this provides insight into how atypical a representation is with respect to the model’s natural computation.
>
> - Detecting dormant behavioral changes: A practical approach is to evaluate interventions across a large and diverse evaluation set, making dormant vulnerabilities less likely to remain undetected.
>
> More generally, we advocate incorporating divergence as a routinely reported metric in causal intervention studies, analogous to how robustness metrics are routinely reported in other subfields.
>
> **Q3. Dormant behavioral changes and the role of CL loss**
>
> We agree that dormant behavioral changes are concerning, and that detecting them is challenging. Our current OOD error analysis does not directly address context-sensitive vulnerabilities because the synthetic setting is not context-dependent.
>
> However, we observe that the CL loss reduces both:
>
> - OOD error, and
>
> - the magnitude of intervened divergence.
>
> Since dormant behavioral changes are more likely when interventions produce highly atypical or off-manifold representations, a reduction in divergence likely lowers their probability. While this is not a formal guarantee, it suggests that the CL loss may help mitigate (though not eliminate) this risk.

---

### Official Review · Reviewer_Rf6s · 2025-11-05

**Soundness:** 2
**Presentation:** 1
**Contribution:** 3
**Rating:** 4
**Confidence:** 3

**Summary:**

This paper examines distributional divergence in mechanistic interpretability interventions. The authors argue that such interventions can push hidden representations off the model’s natural manifold, leading to pernicious effects such as spurious circuit activations or dormant behaviors. They distinguish harmless from pernicious divergences and propose a modified Counterfactual Latent (CL) loss to constrain interventions toward causal subspaces while maintaining interchange intervention accuracy (IIA). The topic is timely and important, and the paper offers useful illustrative examples and intuitions. However, the treatment remains largely informal, with vague definitions, unstated assumptions, and limited quantitative validation. The work would benefit from clearer formalization, stronger theoretical grounding, and more rigorous empirical evaluation to solidify its contribution to addressing divergent representations from causal interventions.

**Strengths:**

**1. Problem statement is timely and important.**
The field of mechanistic interpretability relies heavily on causal interventions, while systematic study of representational divergence has been lacking. The harmless/pernicious dichotomy offers useful language for describing what practitioners often observe intuitively.

---
**2. Useful negative examples.**
The "balanced subspaces" and ReLU examples are compact and concrete; they effectively illustrate how confirmatory behavior can arise from incorrect or misleading activation pathways.

---
**3. Simple mitigation with plausible effect.**
Restricting the CL penalty to causal subspaces aligns with the intended goal of keeping interventions "on-manifold" with respect to identified causal variables.

**Weaknesses:**

**1. Missing formal statements or guarantees.**
The harmless/pernicious taxonomy is presented through narrative examples rather than formal propositions (Sec. `4.1`–`4.2`, `L213`–`L377`). No formal statements, e.g., theorem or proposition, specifies necessary or sufficient conditions under which divergence is guaranteed harmless (e.g., confined to a null-space) or pernicious (activating hidden pathways).
Key terms such as *"behaviorally binary subspace"* (Eq. `4`, `L248`–`L254`) and *"behavioral null-space"* (`L141` and many later references) are invoked but never **formally defined** or tied to measurable properties (e.g., local Jacobians or decision-region invariance). As a result, these claims remain intuitive or conceptual rather than analytical.

---

**2. Under-specified core objects.**
*(1) Behaviorally binary subspace:* The "behaviorally binary subspace" (Sec. `4.1`, Eq. `4`, `L248`–`L254`) is defined via an elementwise sign function mapping to {$-1, 0, 1$}. Including $0$ makes the decision boundary ill-posed: infinitesimal perturbations near $0$ can flip the behavioral label, so the binary characterization becomes unstable unless neighborhoods around $0$ are excluded or measure-zero sets are explicitly treated.

*(2) Null-space:* The "null-space" of downstream layers (Sec. `4.1`, `L213`–`L216`) is repeatedly invoked (e.g., `L481`–`L484`) but never operationally defined under nonlinear composition, normalization, or residual connections. It remains unclear whether this refers to a local Jacobian null-space at $h$ or to a global manifold of invariant directions across contexts.

---

**3. Argument relies heavily on hidden assumptions.**
Several strong but unstated assumptions are embedded in the narrative exposition. For instance, the discussion of "harmless divergence" in Sec. `4.1` presumes a clean factorization between causal subspaces (e.g., $\tilde{\mathbf z}_{var_a}, \tilde{\mathbf z}_{var_b}$) and their covariance structure. This effectively assumes approximate orthogonality and cross-context stability of these subspaces, which is rarely satisfied in deep, nonlinear architectures such as Transformers. Additionally, the notion of harmless covariance within decision boundaries presumes stable, convex class regions (e.g., assumptions and suppositions from `L246`–`L269`), yet the later "dormant behavioral changes" example (Sec. `4.2`) admits that minor contextual shifts can cross those boundaries. The conditions distinguishing truly benign intra-boundary covariance from merely undetected behavioral drift are left unstated. Such implicit assumptions substantially weaken the generality of the reasoning.

---

**4. Evaluation concerns.**
Divergence is visualized using 2D PCA and summarized by a single EMD score (e.g., Fig. `2`). The reviwer concerns whether such low-dimensional projections may distort pairwise distances, and averaging in EMD may mask high-magnitude divergence along select axes.  Additionally, subtracting the baseline EMD of $h$ against itself is reasonable, but since EMD is not linear, simple subtraction might obscure axis-specific drift. Have the authors explored distributional or coordinate-free divergence metrics, reporting with confidence intervals estimated across random projections?

**Questions:**

Please refer to the `Weaknesses` section for detailed questions.

---

The following points are minor and do not impact the overall evaluation:

**1. Notation consistency.**
Throughout the analytic sections (Sec. `4`–`5`), ensure that all notation is defined before use, and that it is applied consistently and unambiguously to improve clarity.

**2. Style.**
Citations should be properly linked, e.g., "Grant (2025)" in the abstract (`L22`). Avoid multiple acronym introductions, such as "causal abstractions (CAs)" (`L126` already introduced, repeated at `L131`). Replace colloquial or vague phrases (e.g., "in causal subspaces is okay", `L280`) with precise language.

---

> ### Author Response · Authors · 2025-11-22
> **Response to assessment**
>
> We thank the reviewer for their thoughtful and constructive assessment.
>
> Below we address each of the weaknesses:
>
> **W1. Missing formal statements or guarantees.**
>
> Thank you for highlighting the need for clearer formal definitions. We have updated the manuscript to increase the precision of the concepts of weight null-space, behavioral null-space, and behaviorally binary subspace.
>
> We now define weight null-space and behavioral null-space at lines 279-283 as follows:
>
> > Formally, we define the null-space of a weight matrix $W\in\mathbb{R}^{d_2\times d_1}$ as $\mathcal{N}(W) = \{v\in\mathbb{R}^{d_1} \,\,| \,\,Wv = 0\}$. Neural activity in the null-space of the weights refers to any changes $\delta\in\mathbb{R}^{d_1}$ for which $W(h^\ell + \delta) = Wh^\ell$. We can generalize this notion beyond the weights to all remaining computations after and including layer $\ell$ in a neural network denoted $f_{\ge \ell}(h)$. We call this the behavioral null-space for a layer $\ell$, defined as $\mathcal{N}(f_{\ge \ell}(h^\ell)) = \{v \in \mathbb{R}^{d_1} \,\, | \,\, f_{\ge \ell}(h^\ell+v) = f_{\ge \ell}(h^\ell)\}$.
>
> And we define a beahviorally binary space on lines 296-304 as follows:
>
> > We can expand this intuition using a \emph{behaviorally binary subspace} as an example, where we define a behaviorally binary subspace as a vector subspace which causally impacts the outputs of a future processing layer (e.g., classification labels) only through its sign.  Formally, let $f:\mathbb{R}^d \to \mathbb{R}^k$ denote a computational layer (possibly consisting of multiple NN layers and functions) where $k$ is the output dimensionality.  Let $\operatorname{sign}(\cdot)$ denote the elementwise sign map $\operatorname{sign}:\mathbb{R}^{d_{\text{var}}} \to \{-1,1\}^{d_{\text{var}}}$, and assume a fixed alignment function $\mathcal{A}$ and subspace selection matrix $D_{\text{var}}\in\mathbb{R}^{d\times d}$.  A linear subspace $Z \subseteq \mathbb{R}^{d}$ is \emph{behaviorally binary (with respect to $f$, $D_{\text{var}}$, and $\mathcal{A}$)} iff for all $D_{\text{var}}\mathcal{A}(h),D_{\text{var}}\mathcal{A}(h') \in Z$,
> > \begin{equation} \operatorname{sign}(D_{\text{var}}\mathcal{A}(h)) = \operatorname{sign}(D_{\text{var}}\mathcal{A}(h')) \;\;\Longrightarrow\;\; f(h) = f(h') \end{equation}
>
> We admit that a limitation of the paper is lack of a clear commitment to a formal definition of the harmless/harmful taxonomy for all causal claims. It is unclear if such a formal definition exists, however. We navigate this issue by stating that harmlessness and perniciousness are not necessarily mutually exclusive on line 053 and 273 and then again at the beginning of section 4.2.
>
>
> **W2. Under-specified core objects.**
>
> In response to the instability of the elementwise sign map, thank you for raising this issue, we agree that instability is an important point. We have removed 0 from the possible output values of the sign map.
>
> In response to the Null-space under nonlinear compositions point, thank you for pointing out the ambiguity. The updated definitions provided in W1 clarify that we are referring to a functional behavioral null-space: invariances under the full computation of the remaining neural network, not just a local Jacobian null-space. This resolves the ambiguity regarding nonlinearities, normalization, and residual paths.
>
>
> **W3. Argument relies heavily on hidden assumptions.**
>
> Yes, in section 4.1 we are assuming that stable, orthogonal causal subspaces exist in some neural architectures. Such subspaces have been shown to exist in synthetic models (Elhage et al. 2022; Geiger et al. 2023) and in LLMs in narrow contexts (Wu et al. 2023). Regardless of whether such subspaces are rare in LLMs, our greater point in Section 4.1 is to show a case in which an interpretability researcher would be indifferent to representational divergence.
>
> **W4. Evaluation concerns.**
>
> Thank you for the opportunity to clarify the point of Section 3 and Figure 2. Our main point in Figure 2 is to demonstrate the existence of distributional differences between the natural and intervened distributions. Thus, a lower bound on the divergence is sufficient to demonstrate our point. We perform the same divergence calculations on the corresponding samples from both the native and intervened distributions, thus, any systematic issues with the divergence metric will be equally applied to both distributions. Also, we note that EMD is computed accross all dimensions in the representations, despite the fact that we only visualize the top principal components (clarified on line 802). We have updated the EMD by normalizing by the vector size which has changed the raw values. We also no longer report the difference in EMD as we noticed that this can obscure the variability of the natural sample. Lastly, we have included additional divergence measures in Appendix A.1.2 and Figure 4 to demonstrate that divergence can be measured in multiple ways.

---

> > ### Author Response · Authors · 2025-11-22
> > **References**
> >
> > Nelson Elhage, Tristan Hume, Catherine Olsson, Nicholas Schiefer, Tom Henighan, Shauna Kravec, Zac Hatfield-Dodds, Robert Lasenby, Dawn Drain, Carol Chen, Roger Grosse, Sam McCandlish, Jared Kaplan, Dario Amodei, Martin Wattenberg, and Christopher Olah. Toy models of superposition. Transformer Circuits Thread, 2022. https://transformer-circuits.pub/2022/toy_model/index.html.
> >
> > Atticus Geiger, Zhengxuan Wu, Christopher Potts, Thomas Icard, and Noah D. Goodman. Finding alignments between interpretable causal variables and distributed neural representations, 2023.
> >
> > Zhengxuan Wu, Atticus Geiger, Thomas Icard, Christopher Potts, and Noah Goodman. Interpretability at scale: Identifying causal mechanisms in alpaca. Advances in neural information processing systems, 36:78205–78226, 2023.

---

> > > ### Comment · Reviewer_Rf6s · 2025-11-26
> > > **Response to authors' rebuttal**
> > >
> > > Thank the authors for their rebuttal. I appreciate the careful reponse to the concerns raised in my previous review. My intial concerns, especially stated in `W4` as well as removing 0 from the sign map outputs in `W2`, are larged addressed. However, some of my key concerns still remains. I further explain them in the following points:
> > >
> > > ---
> > >
> > > 1. The authors add definitions of their main analytic objectives, namely "weight null-space", "behavioral null-space", and "behaviour binary subspace". While these definitions are helpful steps, the rebuttal and the revised manuscript do not fully address the issue that the paper still lacks any formal criteria under when divegence is provably harmless or pernicious. The added definitions do not specify how they determine harmlessness. For example, how the objective is computed or even approximated in a nonlinear, residual-connected network, and under what conditions do perturbations remain in the null-space after multiple layers or under attention mixing? IMHO, the current presentation is not sufficiently analytical, let alone making it a theoretical analysis as highlighted in the title *"addressing divergent representations"* and in the abstract's claim to  *"provide theoretical analysis"*.
> > >
> > > ---
> > >
> > > 2. The authors said in the rebuttal that *"it is unclear if such a formal definition exists"* for the harmless/pernicious taxonomy in response to my `W1`. However, this is precisely my initial concern: the key conceptual claim of the paper is build on a taxonomy that the authors themselves acknowldge may not admit any formalization.
> > >
> > > ---
> > >
> > > 3. In the rebuttal response to my `W2`, the authors claimed that *"This resolves the ambiguity regarding nonlinearities, normalization, and residual paths"*. However, in the definition of behavioral null-space (`L282-283` in the revised manuscript), the set $\mathcal{N}$ is not closed under such cases, which actually can't be a linear subspace, even if we consider only a ReLU layer before an affine map. For example, consider a single layer $f(h) = W \text{ReLU}(h)$ and a hidden state $h$ in the inactive region where $h < 0$ and hence $f(h)=0$. Pick two positive perturbations $v_1, v_2$ that are small enough such that $h+v_1 < 0$ and $h+v_2 < 0$. These perturbations preserve the output, i.e., $f(h+v_1) = 0$ and $f(h+v_2) = 0$. However, if their sum is large enough to cross the activation threshold ($h+v_1+v_2 > 0$), then $\text{ReLU}(h+v_1+v_2) \neq 0$, and generally $f(h+v_1+v_2) \neq 0$. The invariance holds for the components but fails for their sum, which lead the functional invariant set not closed under addition thus can not be a linear subspace. More generally, in a realistic neural network: ReLU regions impose inequality that break linearity; Attention mechanism introduce nonlinear multiplicative invariances; and compositions of nonlinearities will produce non-convex invariance sets. Thus, the authors's use of linear subspace language is not convincing, unless the authors explicitly assume local linearization, justify these assumptions as analytical tools, and acknowledge the analysis-practice gap -- which should be a better manner for an analytical/theoretical paper.
> > >
> > > ---
> > >
> > > As a result, I'd maintain my original evaluation. I do acknowledge the paper identifies an important and under-discussed problem, but the analytical framework still lacks the clarity and rigor to support their central claims. If I'm misunderstanding anything about the content, please correct me.

---

> ### Author Response · Authors · 2025-11-30
>
> Thank you for engaging so deeply with our work.
>
> **1. No formal definition of harmfulness**
>
> Yes, while we maintain our position that harmfulness is claim dependent, we have made this more explicit in the most recent draft on lines 324-334. We point out that an all-encompassing, formal definition of harmlessness is impossible without first establishing a claim. The excerpt is as follows:
>
> > we are not suggesting that the behavioral null-space encompasses the set of all harmless divergences. We speculate that such an exhaustive set is impossible to enumerate without assuming the superiority of some scientific claims/assumptions over others. For example, take the set of all harmless divergences for a specific claim, then modify the claim to assume it permissible to deviate along a vector direction orthogonal to the harmless set. The modification to the claim also modifies the set of harmless divergences. We also note that behaviorally null divergence is not always harmless. Notably, one could {\it desire} to intervene on the behavioral null-space to causally test that it is indeed null. Thus, we stress that the mechanistic claim for which an intervention is meant to support is important for determining the harmlessness of the divergence.
>
> To be clear, we are not suggesting that all attempts at formalizing the harmless/harmful taxonomy are impossible. We are merely stating that such a taxonomy is impossible to formalize without respect to a specific claim. In an effort to remain agnostic to possible future claims, we resort to our updated Section 4.1.
>
> We have changed the beginning of Section 4.1 to be more concrete about presenting the behavioral null-space as a set of harmless divergences for many functional claims. We've modified Section 4.1 to clearly establish a paradigm that 1. establishes a computational unit $\psi$ that is used to define the behavioral null-space, and 2. defines divergence in the behavioral null-space as harmless for claims that treat $\psi$ as an encapsulated function (i.e. the claims do not specify how $\psi$ performs its sub-computations).
>
> The updated text is as follows (lines 276-294):
>
> > Here we explore a set of cases for which we might consider divergent representations to be harmless to functional claims. First among these are cases in which divergence is bottle-necked into the null-spaces of the next interacting weight matrices. Formally, we define the null-space of a weight matrix $W\in\mathbb{R}^{d'\times d}$ as $\mathcal{N}(W) = \{v\in\mathbb{R}^{d} | Wv = 0 \}$. Neural activity in the null-space of the weights refers to any changes $\delta\in\mathbb{R}^{d}$ for which $W(h + \delta) = Wh$. We propose that divergence $v\in\mathcal{N}(W)$ is harmless to the computation of $W$ because it is equivalent to adding the zero vector $W(h+v)=Wh+Wv=Wh+0=W(h+0)$. Notably, however, this harmlessness does not apply to the sub-computations of the matrix multiplication because $v\in\mathcal{N}(W)$ does not imply that $W_{i,j}(h_j+v_j) = W_{i,j}(h_j+0)$ for vector row $j$ and matrix row $i$. Thus, $v$ in this case is potentially harmful to mechanistic claims about individual activation-weight sub-computations, while being harmless to the overall matrix multiplication.
>
> > We can generalize this notion beyond matrices to an arbitrary function $\psi$. Let $\psi:\mathbb{R}^d \to \mathbb{R}^{d'}$ and let $X \subseteq \mathbb{R}^d$. We define the behavioral null-space with respect to $X$ as
> \begin{equation}
> \mathcal{N}(\psi, X)
>     = \{ v \in \mathbb{R}^{d} \mid
>         \forall x \in X,
>         \psi(x+v) = \psi(x)
>     \}.
> \end{equation}
> A common case of $\psi$ in practice for a layer $\ell$ of an NN $f$ consists of all subsequent computations after and including layer $\ell$, denoted $f_{\ge \ell}(h)$. We propose that behaviorally null divergence $v$ is harmless to the overall computation of $f_{\ge \ell}$ because it is equivalent to adding 0 to the input. However, $v$ can be harmful to claims about a sublayer $\ell+k$ within $f_{\ge \ell}$ because $f_{\ell+k}(f_{\ge \ell,< \ell+k}(h+v))$ is not guaranteed to be equal to $f_{\ell+k}(f_{\ge \ell, < \ell+k}(h+0))$ (Sec. 4.2.1). See Appx.A.3 and Algorithm 1 to practically classify harmlessness when $\mathcal{N}(\psi,X)$ characterizes the full space of harmless divergence.
>
>
> **2. Taxonomy without formalization**
>
> See our response to **1.**

---

> ### Author Response · Authors · 2025-11-30
>
> **3. Behaviorally null linear subspaces**
>
> To clarify our definition of a behavioral null-space: we allow the computational unit for which the behavioral null-space is defined to be an arbitrary function. We have hopefully made this more clear with the following excerpt from line 287, “We can generalize this notion [of a null-space] beyond matrices to an arbitrary function $\psi$.”
>
> This means that the behavioral null-space can be a vector subspace for cases in which $\psi(x) = Wx$ where $W$ is a weight matrix, because the behavioral null-space would simply be $\mathcal{N}(\psi,X) = \mathcal{N}(W)$. Even in the example that you provided, it is possible for $v_1$ and $v_2$ to exist in $\mathcal{N}(f,H)$ if the respective column of $W$ consists of all zeros.
>
> If you are pointing out that the behavioral null-space is likely to be defined by complex manifolds (e.g. not linear subspaces), yes, this is possible (and probably common). Our updated Section 4.1 and Equation 4 make it clear that we are defining the behavioral null-space as a set of points without assuming it to exclusively consist of a vector subspace. It is merely the set of points conditioned on $\psi$ and $h$ that do not influence behavior.
>
>
> Thank you again for taking the time to respond!

---

### Official Review · Reviewer_PwXp · 2025-11-05

**Soundness:** 3
**Presentation:** 3
**Contribution:** 3
**Rating:** 8
**Confidence:** 3

**Summary:**

The paper studies mechanistic interpretability approaches that manipulate model representations via targeted interventions and what could go wrong if those interventions result in out-of-domain values that cause unexpected behavior in the model. The paper shows that these divergences are quite common and differentiates them between “harmless” divergences and “pernicious” divergences. A new objective based on counterfactual latent loss is presented to mitigate these issues.

**Strengths:**

The paper studies mechanistic interpretability approaches that manipulate model representations via targeted interventions and what could go wrong if those interventions result in out-of-domain values that cause unexpected behavior in the model. The paper shows that these divergences are quite common and differentiates them between “harmless” divergences and “pernicious” divergences. A new objective based on counterfactual latent loss is presented to mitigate these issues.

**Weaknesses:**

1. The paper addresses an important issue in mechanistic interpretability that is often not considered. If the data that was used to train a model is limited, then the model may be poorly defined on unseen interventions. The interpretability of such cases are therefore potentially unreliable, and this should be taken into account.

2. The points are clearly established with examples, improving the clarity of the paper.

3. The CL loss provides a potential solution to mitigating this issue.

**Questions:**

4. The distinction between harmless and pernicious cases are not well defined, and the paper only provides examples of each, rather than providing a concrete definition of what a divergent representation is, and whether it is harmless or pernicious.

---

> ### Author Response · Authors · 2025-11-22
> **Response to Assessment**
>
> We thank the reviewer for their positive assessment.
>
> In response to the weaknesses, we believe each of the points to reflect positively on the submission. Thank you for noticing these merits.
>
> In response to Question 4. we have now formally defined "divergent representations" at the beginning of Section 4.2 on lines 344-347 as representations that exist outside of the support of naturally occuring representations.
>
> An important point of the paper is that the harmless vs. pernicious classification is partially dependent on the claims for which the experiment is meant to support. This obfuscates the pursuit of making a clear classification boundary. However, we do provide an example of the existence of dormant behavioral changes in now section 4.2.2 (previously 4.2.3), which are a way to solidfy one specific notion of pernicious divergence. Namely, a form of pernicious divergence occurs when the intervened representations are context sensitive.

---

### Official Review · Reviewer_6Hkv · 2025-11-10

**Soundness:** 2
**Presentation:** 2
**Contribution:** 2
**Rating:** 4
**Confidence:** 3

**Summary:**

This paper investigates whether causal interventions in mechanistic interpretability create out-of-distribution (divergent) representations that compromise the faithfulness of resulting explanations. The authors: (1) empirically demonstrate that common intervention methods (mean difference patching, SAEs, DAS) produce divergent representations; (2) provide theoretical analysis distinguishing "harmless" divergences (null-space, within-boundary covariance) from "pernicious" divergences (hidden pathways, dormant behavioral changes); (3) propose adapting the Counterfactual Latent (CL) loss to mitigate divergence. The work is motivated by the fundamental assumption in interpretability that counterfactual model states should be realistic, yet experiments are limited to synthetic settings.

**Strengths:**

1. The paper addresses an important yet underexplored question about whether causal interventions preserve distributional faithfulness, a property essential for claims of mechanistic interpretability.
2. Section 3 systematically examines divergences arising from three distinct intervention methods (mean difference vectors, SAEs, and DAS), demonstrating that the problem is both broad and pervasive.
3. The “harmless vs. pernicious divergence” framework introduced in Section 4 offers a clear conceptual vocabulary for distinguishing when distributional shifts are consequential.
4. The paper is well-structured and logically coherent, progressing naturally from Problem Definition to Empirical Evidence (Section 3), Theoretical Analysis (Section 4), and finally to the Proposed Solution (Section 5).

**Weaknesses:**

1. The proposed solution (CL Loss) is only validated on a synthetic 2D task. However, the problem itself is demonstrated on large models (Llama-3-8B). The gap between these synthetic toy problems and real transformers is enormous. This complete lack of validation on actual LLMs models undermines the practical applicability and credibility of the proposed solution.

2. The "pernicious" case in Example I (Sec 4.2.1) assumes "balanced subspaces" where $w_1h_1 = -w_2h_2$ always holds. If this were true, this subspace would be in the behavioral null-space (its net contribution is always zero) and should not be identified as "causal" by any reasonable method in the first place. The example therefore demonstrates "intervening on wrong variables fails" rather than "divergence on causal variables is pernicious." This conflates choosing incorrect intervention targets with distributional divergence problems, undermining the credibility of the pernicious divergence category.

3. The example in Sec 4.2.2 requires precisely engineered weight matrices, ReLU boundaries, and specific data distributions. The probability of such a fragile and exact configuration occurring in a real, trained network is vanishingly small. This makes the example feel practically irrelevant and weakens the argument that "pernicious" divergence is a common, real-world problem.

4. Even on the simple toy task where the solution should work perfectly, the CL Loss provides minimal benefits. The results show that while EMD (divergence) was reduced, the IIA (task accuracy) actually dropped from 1.0 to 0.998. This suggests the method is weak, ineffective, and may even introduce negative side effects.

**Questions:**

Q1. Real model validation (relates to W1, W3): Can you provide results on at least one real LLM (e.g., GPT-2 Small) showing that CL loss maintains or improves causal identification accuracy while reducing divergence? Even preliminary results would significantly strengthen the paper's claims. In Figure 2(a), more sophisticated causal methods (DAS, SAE) show 2-4× higher divergence than mean-difference patching. Do these methods also achieve higher counterfactual accuracy?

Q2. Example I justification (relates to W2): In Example I, the balanced subspace contributes zero to outputs under natural conditions (as noted in lines 279-280). Can you clarify what makes intervention on such dimensions a case of "pernicious divergence" rather than simply "intervening on wrong (non-causal) variables"? How is this different from general causal identification errors?

Q4. Practical detection (relates to W4): Beyond collecting large evaluation datasets, can you suggest even a heuristic method for practitioners to detect when divergence is likely to be pernicious versus harmless in their specific applications?

---

> ### Author Response · Authors · 2025-11-22
> **Response to weaknesses**
>
> We thank the reviewer for their thoughtful and constructive feedback. We found the comments extremely helpful, and they directly motivated several substantial improvements to the manuscript. Below we address each weakness in detail.
>
> **W1. Lack of real-model validation for the CL loss**
>
> We agree that validation on a real LLM is important. In the revised submission, we added Section 5.1, where we apply the CL loss in a Boundless DAS setup using the experimental configuration of Wu et al. (2023). This experiment uses a Llama-based 7B-parameter model on a practical price-tagging task and evaluates a range of CL-loss weightings. The results show that we can meaningfully reduce divergence while maintaining IIA, providing a concrete proof of principle that the CL loss is feasible and effective in large-scale models.
>
> We also wish to emphasize that—-even absent a proposed solution—-the core phenomenon of representational divergence is sufficiently consequential for causal interpretability that documenting it rigorously is, in our view, a valuable contribution in its own right. The CL loss is intended as an initial, illustrative step toward more robust solutions. We hope its limitations do not detract from the rest of the paper.
>
> **W2. Interpretation of Example I (“balanced subspaces”)**
>
> Thank you for this insightful comment. Your interpretation aligns with one of the key messages we intended to convey. Section 4.2.1 (and Section 4.2 more broadly) aims to show that common causal-interpretability methods can identify subspaces/mechanisms that are behaviorally non-causal under the natural distribution but are causal under interventions. Thus, an intervention cannot always be considered "successful" when it produces experimentally affirming behavior. In Example I, the balanced subspace is indeed behaviorally null under natural conditions—but not under intervened ones. The example highlights how “intervening on the wrong variables” can appear to succeed solely because the intervention has left the natural-distribution support.
>
> Importantly, in the revised submission we also recognize that such perfectly balanced subspaces are unlikely to arise in practical LLMs, as they require two or more rows (or columns) of a weight matrix to be scalar multiples. We verified empirically that this structure does not occur in GPT-2 or Llama-3-8B. Accordingly, we have moved this example to Appendix A.1.4 with added commentary explaining why it is primarily of theoretical interest.
>
>
> **W3. Practicality of Example II (fragility of the configuration)**
>
> We agree that the example in Section 4.2.2 (now Section 4.2.1) is intentionally simplified. It is offered as a pedagogical existence proof, not as a realistic depiction of trained transformer geometry. We chose mean-difference interventions because they are widely used, simple, and the construction illustrates how seemingly successful interventions can activate “hidden pathways”—units, directions, or subcircuits that are inactive on the natural distribution but become active under intervention.
>
> To clarify this mechanism, we have added more precise definitions to Section 4. These formal definitions can be summarized as:
>
> - representational divergence: movement outside the support of natural representations;
> - hidden pathways: components inactive in natural contexts but activated by divergent interventions (units, directions, or subcircuits that are inactive on the natural distribution but become active under intervention).
>
> This improved precision strengthens the argument that such issues may arise in principle even when real networks' exact configurations differ from the constructed example.
>
> **W4. CL loss improvements appear small on the toy task**
>
> We appreciate the reviewer’s concern. In the revision, we added Section 3.1, which formally shows that representational divergence is guaranteed to arise under coordinate patching when manifolds are not axis-aligned hyperrectangles and enough samples are drawn. This helps explain why the CL loss cannot achieve perfect mitigation when restricted to orthogonal alignment.
>
> To address this, we relaxed the alignment constraint from strict orthogonality to general linear invertibility. Under this setting:
>
> - the model now achieves 1.0 accuracy on the previously reported task;
> - on a new, more rigorous train/test split with held-out classes, CL loss outperforms DAS alone (0.998 vs. 0.996).
>
> We also emphasize that the CL loss is intended as a supplement to causal interventions rather than a standalone replacement.
>
> **References**
> Zhengxuan Wu, Atticus Geiger, Thomas Icard, Christopher Potts, and Noah Goodman. Interpretability at scale: Identifying causal mechanisms in alpaca. Advances in neural information processing systems, 36:78205–78226, 2023.

---

> > ### Author Response · Authors · 2025-11-22
> > **Response to Questions**
> >
> > **Q1. Validation on real LLMs**
> >
> > Yes. As noted above, we have added Section 5.1, providing results from a Llama-based Alpaca 7B model. These results directly respond to both Q1 and W1.
> >
> > Regarding the question on counterfactual accuracy across methods:
> > Because the methods in Figure 2 are applied to different tasks designed to probe different structural questions, direct comparisons of counterfactual accuracy are potentially not meaningful. Nonetheless:
> >
> > - mean-difference patching reaches 1.0 accuracy,
> >
> > - DAS achieves 0.96–0.97,
> >
> > - SAEs were not evaluated in a behavioral-accuracy context.
> >
> > **Q2. Why Example I counts as “pernicious divergence”**
> >
> > See our response to W2. Briefly: the point was that representational divergence can cause otherwise reasonable causal-identification methods to misattribute causality, even when the underlying mechanism is null on the natural distribution. Since this example is unlikely in practice, we have moved it to the appendix and clarified its interpretive purpose.
> >
> > **Q4. Detecting pernicious divergence in practice**
> >
> > We agree that practical detection is important. The most direct heuristic is to evaluate interventions on out-of-distribution contexts, where hidden pathways are most likely to fail to produce affirming behavior. However, the notion of “harm” is inherently claim-dependent: perniciousness depends on what causal conclusion the practitioner intends to draw. We have added lines 272-273, "we stress that the harm is inherently claim dependent, meaning that these divergences are not mutually exclusive."
> >
> > To provide actionable guidance, we added the following recommendation (lines 414–416):
> >
> > > “Causal-intervention experiments should ideally (1) report any representational divergence introduced outside the null-space, and (2) test whether interventions exhibit context-sensitivity across distributions.”

---

> > > ### Comment · Reviewer_6Hkv · 2025-11-26
> > >
> > > I appreciate the authors’ responses and clarifications. However, I still find it difficult to assess the practical utility of the method, including its scalability to real models and datasets and its concrete benefits in realistic settings. I will maintain my current score.

---

> > > > ### Author Response · Authors · 2025-11-30
> > > >
> > > > The practical utility of the paper is as follows:
> > > >
> > > > - it theoretically and empirically demonstrates a widespread phenomenon that is often unaddressed
> > > > - it explores ways of formalizing the phenomenon in an effort to ensure rigorous NN interpretability
> > > > - it raises mechanistic proofs of principle for potential shortcomings of existing causal interpretability methods
> > > > - it presents a mitigating solution to the problem for a widely used interpretability method in a practical setting
> > > > - it presents an entirely new interpretability method on synthetic tasks
> > > >
> > > > Furthermore, in our most recent draft, we have included one final analysis showing an inverse correlation between divergence and OOD intervention accuracy on lines 519-521. This demonstrates that divergence can be practically used as a predictor for the OOD generalization of a given intervention. And we have included an additional algorithm 1 in Appendix A.3 for determining harmless divergence for cases in which the behavioral null-space characterizes the full harmless set.
> > > >
> > > > We appreciate your efforts as a reviewer!

---

### Author Response · Authors · 2025-11-22
**Thank you for encouraging multiple improvements to the paper**

We thank all reviewers for their time, thoughtful feedback, and constructive suggestions. The reviews substantially improved the clarity and rigor of the paper. In response, we have uploaded a revised version of the manuscript that incorporates several significant additions and refinements.

The major changes are summarized below:

- *New theoretical subsection (Section 3.1).* We added a formal result showing that, for most activation manifolds, there exist source–target vector pairs that produce representational divergence under simple patching operations. This strengthens the argument in Section 3 that representational divergence is a widespread and generic phenomenon.

- *Clarified definitions and terminology (Section 4).* We improved the precision of our definitions of null spaces (Section 4.1), hidden pathways (Section 4.2.1), and dormant behavioral changes (Section 4.2.2), etc. addressing multiple reviewer requests for clearer formalism.

- *New Llama-7B experiment demonstrating practical applicability (Section 5.1).* We added an experiment applying the CL loss in a Boundless DAS setup using a 7B-parameter Llama-based model on a practical price-tagging task. This provides a concrete proof of principle that the CL loss can be efficiently deployed in large-scale, real-world settings.

- *Expanded divergence analyses (Figure 4).* We included several additional divergence metrics to provide a more comprehensive empirical characterization.

---

### Author Response · Authors · 2025-12-02
**Review/Rebuttal Discussion Summary**

Summary of the criticisms and our responses:

- **Lack of real world validation for CL loss:** Many of the reviewers said that they could not evaluate the practical value of the mitigating solution offered in the original Section 5 (i.e. the modified CL loss in a synthetic setting).
    - In response, we have added an experiment using the CL loss in a practical, price-tagging task on a Llama based 7B parameter LLM (Sec. 5.1) using the experimental setting from Wu et al. 2023. Figure 3B shows that the CL loss can reduce divergence while maintaining interpretive power.

&nbsp;

- **Lack of practical taxonomy for harmless vs harmful divergences:** Reviewers raised the concern that although the paper demonstrates that intervened divergence exists, and it demonstrates that divergence can be both harmless and pernicious, the paper fails to provide a practical guide on how to classify harmless vs. pernicious divergences.
    - We maintain our initial position that a complete formal taxonomy cannot exist independently of a set of corresponding scientific claims
    - We have also added a practical algorithm (Algorithm 1) to Appendix A.3 for classifying harmless divergence in cases where the behavioral null-space (defined in equation 4) characterizes the full set of harmless divergences.

&nbsp;

- **Lack of formalization of important terms:** Two reviewers (PwXp and Rf6s) said that important terms were under defined.
    - we have improved the formalization of the behavioral null-space, behaviorally binary subspaces, hidden pathways, dormant behavioral changes, and divergent representations

&nbsp;

- **Examples require strong assumptions in Section 4.2:** Two reviewers (Rf6s and Ahsj) said that the examples in Section 4 require strong assumptions. One reviewer (6Hkv) said that the exact example in updated Section 4.2.1 is unlikely to exist in real models.
    - The _balanced subspaces_ example (prev Sec 4.2.1) was moved to the appendix (noting the core illustration is unlikely to exist in real models).
    - The _mean difference vector patching_ example (updated Sec 4.2.1) is meant as a proof of principle of how pernicious divergence can occur mechanistically even from a relatively simple causal method. It also shows that experimentally affirming behavior does not necessarily support an understanding of the NN’s mechanistic solution.

&nbsp;

- **Divergence evaluation concerns:** Reviewer Rf6s suggested that Earth Mover’s Distance (EMD) to measure divergence is subject to masking high divergence along select axes.
    - We have included Figure 4 showing 7 more divergence metrics. Each shows a similar story defending Section 3. Also, any estimation errors are applied equally to both the natural and intervened distributions.

---

> ### Author Response · Authors · 2025-12-02
> **Summary of Most Recent Manuscript Changes**
>
> We wish to highlight that we have updated the manuscript after the policy change that has restricted reviewer responses. The changes are summarized as follows. We have:
>
> - added a link to the anonymized, public code for reproducibility
> - improved the clarity and formality of behavioral null-spaces in Section 4.1
> - added clear, explicit language demonstrating the impossibility of a formal taxonomy distinguishing harmlessness/perniciousness without reference to a specific claim (end of Section 4.1)
> - added Appendix A.3 detailing a practical algorithm for approximating the harmlessness of divergence when the behavioral null-space fully characterizes harmlessness
> - provided an analysis showing that EMD along causal-subspaces in the training data negatively predicts OOD intervention performance on the evaluation set, thus improving the practical value of Section 5 (end of Section 5.2).

---

### Meta-Review · Area_Chair_nQZu · 2025-12-07

**Summary:**

This study proposes and investigates notions of "divergence" in counterfactual interventions to neural networks. Specifically, what is divergence, when is it harmless, and when might it affect the claims made by studies that deploy interventional techniques (so-called "pernicious divergence")? Reviewers largely agreed that this is an idea that would directly impact a great proportion of contemporary interpretability research.

Reviewers' concerns largely focused on (1) the actual prevalence of harmless vs. pernicious divergence in less artificial experimental settings, (2) the practical utility of the proposed CL loss solution in real settings, and (3) unclear definitions and assumptions. While I agree with reviewers that it is not yet totally convincing that the proposed ideas and solutions would apply in realistic settings, I believe that the proofs-of-existence are sufficiently convincing to make me agree with the paper's main recommendation—that researchers in this area should be engaging with divergence more directly and frequently than is currently the case. The authors have done a great job responding to (3), in my opinion.

Thus, while there are still some outstanding concerns regarding applicability to real-world settings, I believe that the most important details and claims are now clear and sound, and that the main ideas in this paper are likely to be impactful to a broad audience.

**Reviewer Concerns:**

1. The actual prevalence of the two types of divergence is still in question. The proofs-of-existence are quite artificial, although the authors have added some evidence of this phenomenon existing in real language models. I believe this has been >50% addressed, but not 100% addressed.

2. The concern about the practical utility of the proposed CL loss is still outstanding. It was argued that one might want to relax the assumption that any divergence is bad, as this larger solution space could lead to more powerful intervention techniques. I agree with this, but also believe that this weakness was not particularly major given that the new experiment does show benefits in full-scale models. (The authors acknowledge that this is more of a first-pass existence demonstration that divergence can be mitigated, and that this is not meant to be a final, definitive, nor even principled solution to this problem.)

3. I believe that points relating to unclear formalizations and assumptions have been very well addressed during discussions with Reviewer Rf6s.

**Reviewer Scores:**

Reviewer Ahsj has explicitly stated that they had increased their score to an accept. Two out of four weaknesses raised by Reviewer Rf6s have been addressed, according to them. Reviewer 6Hkv is still skeptical of the practical utility of the proposed solution, despite additional experiments with full-scale language models on a more realistic task. It is possible that YbiP may have increased their scores, as the authors have added experiments directly addressing these concerns, but the evidence here was not convincing to another reviewer with similar concerns; it is therefore difficult to say what they would have done.

---

### Decision · Program_Chairs · 2026-01-26

Accept (Oral)